

# Exact solution for the Lindbladian dynamics for the open XX spin chain with boundary dissipation

**Kohei Yamanaka⋆ and Tomohiro Sasamoto**

Department of Physics, Tokyo Institute of Technology,
Ookayama 2-12-1, Tokyo 152-8551, Japan

⋆ yamanaka@stat.phys.titech.ac.jp

## Abstract

We obtain exact formulas for the time-dependence of a few physical observables for the open XX spin chain with Lindbladian dynamics. Our analysis is based on the fact that the Lindblad equation for an arbitrary open quadratic system of $N$ fermions is explicitly solved in terms of diagonalization of a $4N \times 4N$ matrix called structure matrix by following the scheme of the third quantization. We mainly focus on the time-dependence of magnetization and spin current. As a short-time behavior at a given site, we observe the plateau regime except near the center of the chain. Basic features of this are explained by the light-cone structure created by propagations of boundary effects from the initial time, but we can explain their more detailed properties analytically using our exact formulas. On the other hand, after the plateau regime, the magnetization and spin current exhibit a slow decay to the steady state values described by the Liouvillian gap. We analytically establish its $O(N^{-3})$ scaling and also determine its coefficient.



# 1   Introduction

Open non-equilibrium systems, connected with external reservoirs, have been one of the most important subjects in non-equilibrium statistical mechanics [1,2]. They are known to show various interesting behaviors and phenomena, which are not seen in systems in thermal equilibrium. A classical example is the Bernard convection, in which a characteristic spatio-temporal pattern appears when the temperature difference between the top and bottom sides of an intermediate liquid becomes large enough [3–5]. To understand basic properties of non-equilibrium systems, studying simple model systems is useful. In particular, there have been extensive studies on classical one-dimensional models which show nontrivial phenomena like boundary induced phase transition and anomalous transport and at the same time are analytically tractable [6–8].

Recently, due to the development of experimental techniques, non-equilibrium states are realized also in a variety of quantum systems, such as cold atoms [9–12], optics [13, 14], and quantum walks [15]. Correspondingly studying non-equilibrium properties of open quantum systems from a theoretical point of view is also becoming more and more important. In addition, for the last few years, connections to studies of non-hermitian systems have been suggested and attracted attention [16–21], since open quantum systems can be interpreted as non-hermitian systems.

There are a few theoretical frameworks to study the dynamics of open quantum systems. A conventional one is the use of non-equilibrium Green's function [22,23], which is an extension of the standard Green's function [23, 24] and has been useful to analytically calculate time dependent correlation functions for systems in equilibrium. Recently, the method has been generalized to study systems in which the state evolves from a given initial condition to another [25,26]. It has been already applied to a few concrete models such as the one-dimensional XY spin chain [27–29]. In this approach, the time evolution is still given by a Hamiltonian, but calculations tend to be rather cumbersome. It has turned out that a description by a quantum master equation [30–32] is equally effective and useful to study various properties of non-equilibrium systems. There are several versions of the quantum master equations, such as the Lindblad equation and the Redfield equation. In this paper we employ the description by the Lindblad equation. We remark that relationships between the quantum master equations and the method of the non-equilibrium Green's function have been recently examined [33].

The Lindblad equation has been mainly solved numerically, by which one can treat only relatively small systems. But by taking simple models which are analytically tractable, we may study non-equilibrium properties of large systems. Indeed there have been already some previous works for several one-dimensional systems described by the Lindblad equation. In particular, a few exact solutions for the nonequilibrium steady states(NESSs) have been obtained by using Matrix Product Ansatz (MPA) [34–42]. As for dynamics, there has been some recent progress in numerical calculations such as the Matrix Product Operator method [43–45], the

density matrix renormalization group method [46,47]. It is equally important to develop analytical techniques to study their dynamics [48–50]. In particular, analytical solutions for some simple model systems would provide invaluable information for understanding general open quantum systems.

In this paper, we will give an exact solution for the time-dependence of the magnetization and the spin current for the XX spin chain with boundary dissipation described by the Lindblad equation. We will use the fact that an arbitrary open quadratic system whose dynamics is described by the Lindblad equation admits an application of the third quantization [51]. Although this method has been already known for about ten years and has been applied to several fermionic and bosonic systems [51–57], as far as we know, it has not been fully exploited for obtaining exact formulas for time-dependent physical quantities. In this paper we will show how we can utilize the third quantization to obtain exact time-dependence of physical quantities and provide explicit formulas for a few of them.

In previous works [51, 53–57], solving a Lindblad equation describing the dynamics for open quadratic bosonic/fermionic systems has been shown to reduce to a diagonalization of a $2N \times 2N$ matrix. In this paper, we show that, in the case of the open XX spin chain, the problem can be further reduced to a diagonalization of an $N \times N$ non-Hermitian matrix and that this non-Hermitian matrix can be diagonalizable. We remark on the fact that solving a Lindblad equation describing the dynamics has been shown to reduce a diagonalization of the $N \times N$ non-Hermitian matrix in a few specific cases, such as for the open XX spin chain whose specific dissipative strengths satisfy the condition $4J^2 = \varepsilon_L \varepsilon_R$ [56], and the open XY spin chain without magnetic field [57]. Using our procedure, the non-Hermitian matrix for the open XX spin chain can be diagonalized for arbitrary dissipative strengths and magnetic field. Then we will show that the time-dependence of physical quantities can be studied by solving the continuous-time differential Lyapunov equation [58–60] and that this equation can indeed be solvable. By combining these we can arrive at explicit formulas for the time-dependence for an open quantum system described by the Lindblad equation for the first time. We also remark that a similar reduction of matrix size has been known for the XY spin chain Hamiltonian in the context of the Kitaev model [61, 62].

As an example of applications of our formulas, we consider the time-dependence of the magnetization and the spin current from the thermal equilibrium state in the high temperature limit $\beta \to 0$. First, by taking the limit $t \to \infty$, we obtain the exact solutions for the NESS. We will see that our formulas give a generalization of the formulas in a previous study using MPA [35], in which only the case of opposite magnetizations at the boundaries was treated. By the same formulas, we will also analyze behaviors for time-dependent physical observables. We first observe that the spatio-temporal dependence of the magnetization for the open XX chain using our formulas shows a light-cone structure. Similar light-cone structures have appeared in quench dynamics or a dynamics starting from the step initial condition [11,63,64]. Our results would be useful to discuss similarities and differences with the dynamics of the closed XX spin chain and the validity of some approximations in the derivation of the QMEs [32]. By carefully examining the behaviors of physical quantities, we can study various other properties as well. For example we can analytically show the emergence of the plateau regime and discuss their behaviors in detail by performing an asymptotic analysis of integral representations of physical quantities. Also, after the plateau regime, we observe a slow relaxation for the magnetization and the spin current at a bulk site, corresponding to the Liouvillian gap. By examining our formulas, we will not only establish the $O(1/N^3)$ scaling but also determine its coefficient.

The paper is organized as follows. In the following section 2, we shortly explain the general theorems of the third quantization to review the previous studies [51, 55], and we calculate the exact spectrum of the Lindbladian. In sections 3 and 4, we explain the main results of this paper. In section 3, we explain: (i) we can calculate the analytical steady state solutions of

the magnetization and spin current for open XX spin chain with left-right asymmetric dissipation strength and bath magnetization, and (ii) the exact solutions of the time-dependence of magnetization and spin current are obtained. In section 4, we focus on several specific behaviors for the dynamics of the open XX spin chain with boundary dissipations. In particular, we analytically discuss the light-cone structure, the plateau regime where the magnetization does not change over a duration of time, and the Liouvillian gap. The former two issues appear in a short time window, and the latter one is related to a long time window. Each time window is determined by the specific time for this system, and we introduce these in section 4. In section 5, we summarize this paper, and in appendixes we give more detailed calculations for the physical observables for steady state and the time-derivative of the magnetization on an arbitrary site.

## 2 Spectrum of the open XX spin chain with boundary dissipation

### 2.1 Lindbladian in Liouvillian-Fock space

We consider the following Hamiltonian of XX spin chain,

$$H = J \sum_{k=1}^{N-1} (\sigma_k^x \sigma_{k+1}^x + \sigma_k^y \sigma_{k+1}^y) - B \sum_{k=1}^{N} \sigma_k^z, \tag{1}$$

where $\sigma_k^{x,y,z}$ are the Pauli operators, $J$ is the coupling constant between a site and nearest-neighbor sites, and $B$ is denoted as the magnetic field. The Lindblad equation [31] is denoted as

$$\frac{\mathrm{d}}{\mathrm{d}t}\rho(t) \equiv \mathcal{L}\rho(t) = -i[H,\rho(t)] + \sum_{\mu} L_\mu \rho(t) L_\mu^\dagger - \frac{1}{2}\left\{L_\mu^\dagger L_\mu, \rho(t)\right\}, \tag{2}$$

where $\rho(t)$ is the density operator and Lindblad dissipative operators are defined as

$$L_1 = \sqrt{\varepsilon_{\mathrm{L}}\frac{1+\mu_{\mathrm{L}}}{2}}\sigma_1^+, \quad L_3 = \sqrt{\varepsilon_{\mathrm{R}}\frac{1+\mu_{\mathrm{R}}}{2}}\sigma_N^+, \tag{3}$$

$$L_2 = \sqrt{\varepsilon_{\mathrm{L}}\frac{1-\mu_{\mathrm{L}}}{2}}\sigma_1^-, \quad L_4 = \sqrt{\varepsilon_{\mathrm{R}}\frac{1-\mu_{\mathrm{R}}}{2}}\sigma_N^-, \tag{4}$$

where $\sigma^\pm = (\sigma^x \pm i\sigma^y)/2$, $\varepsilon_{\mathrm{L/R}}$ are dissipative strength between the system and each reservoir, and $\mu_{\mathrm{L/R}}$ are the magnetization on each reservoir. We can explain the interpretation of these parameters and the forms of the operators (3,4) when we derive the Lindblad equation from the dynamics of the total system including the reservoirs [65, 66]. The Lindblad operators $L_1, L_2$ (3,4) play the roles of entry and exclusion of the up-spin between the left boundary and the left end, and $L_3, L_4$ (3,4) play the roles of entry and exclusion for the up-spin between the right boundary and the right end. These parameters $\varepsilon_{\mathrm{L/R}}, \mu_{\mathrm{L/R}}$ are related to the coupling strength in each boundary and each reservoir's chemical potential, respectively.

In the following, we will determine the spectrum of the Lindbladian $\mathcal{L}$ in (2), which is a linear operator in the space of density operators. A summary will be given at the end of this section.

We introduce the Majorana fermion operators $w_j$, $j = 1, 2, \cdots, 2N$ satisfying the anti-commutation relations $\{w_j, w_k\} = 2\delta_{j,k}$. The XX spin chain is equivalent to the one-dimensional free Majorana fermion model using the inverse of the Jordan-Wigner transformation $\sigma \to w$. These operators $w_j$ are related to Pauli operators $\sigma_m$ as the following Jordan-Wigner transformation [51],

$$w_{2k-1} = \sigma_k^x \prod_{n<k} \sigma_n^z, \quad w_{2k} = \sigma_k^y \prod_{n<k} \sigma_n^z, \quad \leq k \leq N. \tag{5}$$

The Hamiltonian in (1) and Lindblad dissipative operators in (3,4) are rewritten in terms of the Majorana fermion operators $w_j$ as

$$H = -iJ \sum_{k=1}^{N-1} (w_{2k} w_{2k+1} - w_{2k-1} w_{2k+2}) + iB \sum_{k=1}^{N} w_{2k-1} w_{2k}, \tag{6}$$

and as

$$L_1 = \sqrt{\varepsilon_{\mathrm{L}} \frac{1+\mu_{\mathrm{L}}}{2}} \frac{w_1 + i w_2}{2}, \qquad L_2 = \sqrt{\varepsilon_{\mathrm{L}} \frac{1-\mu_{\mathrm{L}}}{2}} \frac{w_1 - i w_2}{2}, \tag{7}$$

$$L_3 = \sqrt{\varepsilon_{\mathrm{R}} \frac{1+\mu_{\mathrm{R}}}{2}} \frac{w_{2N-1} + i w_{2N}}{2} \boldsymbol{\Omega}, \quad L_4 = \sqrt{\varepsilon_{\mathrm{R}} \frac{1-\mu_{\mathrm{R}}}{2}} \frac{w_{2N-1} - i w_{2N}}{2} \boldsymbol{\Omega}, \tag{8}$$

respectively. Here, $\boldsymbol{\Omega} := (-1)^N \prod_{l=1}^{2N} w_l$ is a Casimir operator which commutes with all the elements of the Clifford algebra generated by Majorana operators $w_j$, and satisfies $\boldsymbol{\Omega}\boldsymbol{\Omega}^\dagger = \boldsymbol{\Omega}^\dagger \boldsymbol{\Omega} = 1$.

Throughout this paper, $\underline{x} = (x_1, x_2, \cdots)^{\mathrm{T}}$ will designate a vector (column) of appropriate scalar valued or operator valued symbols $x_k$. Then, the Hamiltonian and the Lindblad dissipative operators (6-8) can be expressed by a quadratic form and linear forms respectively as

$$H = \sum_{j,k=1}^{2N} w_j \mathrm{H}_{j,k} w_k = \underline{w} \cdot \mathbf{H} \underline{w}, \tag{9}$$

$$L_\mu = \sum_{j=1}^{2N} l_{\mu,j} w_j = \underline{l}_\mu \cdot \underline{w}, \tag{10}$$

where $\underline{A} \cdot \underline{B}$ is the inner product between the vectors $\underline{A}$ and $\underline{B}$, and $2N \times 2N$ matrix $\mathbf{H}$ can be chosen to be an antisymmetric matrix $\mathbf{H}^{\mathrm{T}} = -\mathbf{H}$. From Lindblad dissipative operators, the matrix $\mathbf{M}$ is defined as

$$\mathrm{M}_{jk} = \sum_\mu l_{\mu,j} l_{\mu,k}^*, \tag{11}$$

which is a Hermitian matrix, and $\mathbf{M}_R$ and $\mathbf{M}_I$ are real and imaginary part of the matrix $\mathbf{M}$, respectively.

A fundamental concept of the third quantization [51] is the Fock structure on $4^N$-dimensional Liouville space of operators $\mathcal{K}$, called the operator space. This space is created as the Hilbert space of density operators with the definition of an inner product $\langle A|B \rangle = 4^{-N} \mathrm{tr}(A^\dagger B)$ where $A, B$ are operators. We use Dirac bra-ket notation for the operator space $\mathcal{K}$. This means replacing the relation between operators and states over physical Hilbert space with the one between maps and operators over the operator space. Then, symbols with a hat shall designate linear maps over the operator space $\mathcal{K}$, and we note the difference between an operator $X$ over the physical Hilbert space and a map $\hat{X}$ over operator space $\mathcal{K}$. By this transformation, the Lindblad equation (2) is rewritten as

$$\frac{\mathrm{d}}{\mathrm{d}t} |\rho(t)\rangle = \hat{\mathcal{L}} |\rho(t)\rangle. \tag{12}$$

The Lindblad map $\hat{\mathcal{L}}$, which may be related to the Lindbradian $\mathcal{L}$ in (2) by a similarity transformation, is written in terms of the self-adjoint Hermitian Majorana fermion maps $\hat{a}_{\mu,r}$ [51] satisfying $\{\hat{a}_{\mu,r}, \hat{a}_{\nu,s}\} = \delta_{\mu,\nu}\delta_{r,s}$, and this map takes a quadratic form with the identity map term $\mathbb{1}$ as

$$\hat{\mathcal{L}} = \underline{\hat{a}} \cdot \mathbf{A}\underline{\hat{a}} - A_0\hat{\mathbb{1}}, \tag{13}$$

where a matrix $\mathbf{A}$ is called the structure matrix

$$\mathbf{A} = \begin{pmatrix} -2i\mathbf{H} + i\mathbf{M}_I & i\mathbf{M} \\ -i\mathbf{M}^{\mathrm{T}} & -2i\mathbf{H} - i\mathbf{M}_I \end{pmatrix}, \tag{14}$$

and the coefficient of identity term $A_0$ is equal to the trace of the matrix $\mathbf{M}$. It is known that eigenvalues and eigenvectors of the Lindblad map $\hat{\mathcal{L}}$ (or Lindbradian $\mathcal{L}$) can be constructed from those of the structure matrix $\mathbf{A}$ [54].

The Lindblad map conserves its parity. The operator space $\mathcal{K}$ can be decomposed into a direct sum $\mathcal{K} = \mathcal{K}_+ \oplus \mathcal{K}_-$ which are defined as $\mathcal{K}_{\pm} = \frac{1 \pm \exp(i\pi\sum_k(\frac{1}{2} - i\hat{a}_{1,k}\hat{a}_{2,k}))}{2}\mathcal{K}$. Then, the parity of the Lindblad map in the operator space $\mathcal{K}$ corresponds to that of total number of the Majorana operator $w_j$ in physical Hilbert space $\mathcal{H}$. In this paper, we consider only the product of an even number of the Majorana fermion operator $w_j$, which is enough to calculate usual physical observables, for example magnetization, spin current, energy, and so on. Thus, we can restrict our attention to the subspace $\mathcal{K}_+$. If the structure matrix $\mathbf{A}$ is written as the Jordan canonical form, the Lindblad map $\hat{\mathcal{L}}$ becomes the almost-diagonal map. Moreover, we obtain the exact solution of the time-dependence of physical observables whose dynamics are described by the Lindblad equation.

## 2.2 Exact Spectrum of Lindbladian

As shown in [54], the structure matrix $\mathbf{A}$ is unitary equivalent to a following block-triangular matrix,

$$\tilde{\mathbf{A}} = \mathbf{U}\mathbf{A}\mathbf{U}^{\dagger} = \begin{pmatrix} -\mathbf{X}^{\mathrm{T}} & 2i\mathbf{M}_I \\ 0 & \mathbf{X} \end{pmatrix}, \tag{15}$$

where $\mathbf{X} = -2i\mathbf{H} + \mathbf{M}_R$ is a real matrix, and the matrix $\mathbf{U}$ is trivially the $4N \times 4N$ permutation matrix which corresponds to the cyclic permutation of Pauli operators $(\sigma^x \to \sigma^y, \sigma^y \to \sigma^z, \sigma^z \to \sigma^x)$. Also, as shown in [54], if the matrix $\mathbf{X}$ is diagonalizable, the structure matrix is diagonalizable. Thus, we consider only the eigensystem of a $2N \times 2N$ matrix $\mathbf{X}$. Moreover it has been known, in the specific cases of the open XX spin chain whose specific dissipative strengths satisfy the condition $4J^2 = \varepsilon_{\mathrm{L}}\varepsilon_{\mathrm{R}}$ [56] and the open XY spin chain without magnetic field [57], that the matrix $\mathbf{X}$ can be decomposed into $N \times N$ matrices. In this paper, we show that, for the open XX spin chain with general magnetic field and dissipative parameters, the matrix $\mathbf{X}$ can be decomposed into $N \times N$ matrices.

**Lemma 1.** Using a unitary matrix $\mathbf{S}$, the matrix $\mathbf{X}$ is unitarily equivalent to a block-diagonal matrix

$$\tilde{\mathbf{X}} = \mathbf{S}\mathbf{X}\mathbf{S}^{\dagger} = \begin{pmatrix} i\Xi & 0 \\ 0 & -i\Xi^{\dagger} \end{pmatrix}, \tag{16}$$

where $\Xi$ is an $N \times N$ matrix.

We can show this lemma easily. First, the matrix $\mathbf{X}$ is rewritten by using the Kronecker product

$$\mathbf{X} = i \begin{pmatrix} B & J & & \\ J & B & & \\ & & \ddots & J \\ & & J & B \end{pmatrix} \otimes \sigma^y + \begin{pmatrix} \frac{\varepsilon_{\mathrm{L}}}{4} & & & \\ & 0 & & \\ & & \ddots & \\ & & & 0 \\ & & & & \frac{\varepsilon_{\mathrm{R}}}{4} \end{pmatrix} \otimes \mathbb{1}_2. \tag{17}$$

Then, we introduce the following permutation,

$$\kappa :\mapsto \left\{ \begin{array}{cccccccc} 1, & 2, & \cdots, & N, & N+1, & \cdots, & 2N-1, & 2N \\ 1, & 3, & \cdots, & 2N-1, & 2, & 4, & \cdots, & 2N \end{array} \right\}. \tag{18}$$

The $2N \times 2N$ permutation matrices which correspond to the above permutation and the cyclic permutation of Pauli operators are defined to be $\mathbf{\Pi}_\kappa$ and $\check{\mathbf{U}}$, and the unitary matrix $\mathbf{S}$ is denoted as $\mathbf{S} = \check{\mathbf{U}}\mathbf{\Pi}_\kappa$. The matrix $\mathbf{X}$ is decomposed into the form of a block matrix as

$$\tilde{\mathbf{X}} = \mathbf{S}\mathbf{X}\mathbf{S}^\dagger = \begin{pmatrix} i\Xi & 0 \\ 0 & -i\Xi^\dagger \end{pmatrix}, \tag{19}$$

where the matrix $\Xi$ is non-Hermitian matrix

$$\Xi = \begin{pmatrix} B - i\frac{\varepsilon_{\mathrm{L}}}{4} & J & & & \\ J & B & & & \\ & & \ddots & & \\ & & & B & J \\ & & & J & B - i\frac{\varepsilon_{\mathrm{R}}}{4} \end{pmatrix}. \tag{20}$$

Also, we can decompose the characteristic polynomial of the matrix $\mathbf{X}$ into two characteristic polynomials of the matrix $\Xi$, since the matrix $\tilde{\mathbf{X}}$ is block-diagonalizable.

**Corollary 1.** The characteristic polynomial of the matrix $\mathbf{X}$ is decomposed into two characteristic polynomials of the matrix $\Xi$

$$p_{\mathbf{X}}(\lambda) = p_\Xi(-i\lambda)p_\Xi^*(-i\lambda^*), \tag{21}$$

where $p_{\mathbf{X}}(\lambda) := \det(\mathbf{X} - \lambda\mathbb{1}_{2N})$, and $p_\Xi(\lambda) := \det(\Xi - \lambda\mathbb{1}_N)$.

Therefore, all the eigenvalues of the Lindblad map $\hat{\mathcal{L}}$ (or Lindbradian $\mathcal{L}$) for the open XX spin chain are constructed by the eigenvalues of the $N \times N$ matrix $\Xi$. Moreover, the matrix $\Xi$ is a tri-diagonal matrix and we can obtain the eigenvalues and eigenvectors of the matrix $\Xi$ [29,67–69]. Consider the eigenvalue problem $\Xi\underline{q} = \lambda\underline{q}$ where the $k$-th$(1 \leq k \leq N)$ eigenvector $\underline{q}^{(k)} = (q_1^{(k)}, q_2^{(k)}, \cdots, q_N^{(k)})^{\mathrm{T}}$. In the following we will set $q_1^{(k)} = 1$, since the value of $q_1^{(k)}$ can be an arbitrary non-zero number. Then, we obtain the eigenvalue and the component of the eigenvector [67,68]

$$\lambda^{(k)} = B + 2J\cos\theta_k, \tag{22}$$

and

$$q_j^{(k)} = \frac{1}{\sin\theta_k}\left[\sin j\theta_k + il\sin(j-1)\theta_k\right], \tag{23}$$

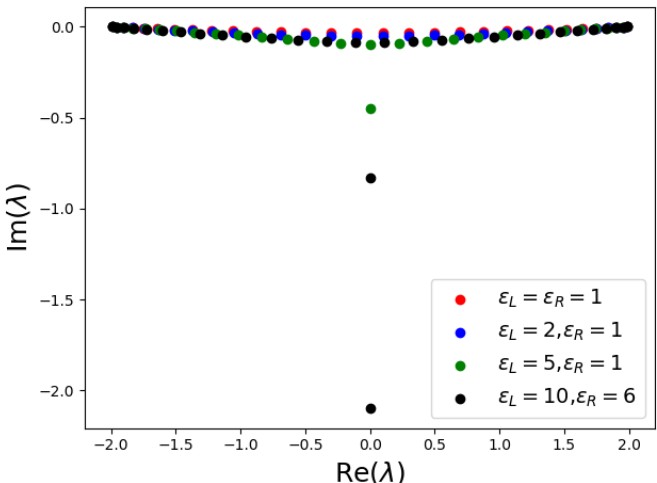

Figure 1: Eigenvalue distribution of matrix $\Xi$. Other parameters are set to $N = 30$, $J = 1.0$, $B = 0.0$, and $\mu_{\mathrm{L}} = -\mu_{\mathrm{R}} = 1.0$.

where the parameter $\theta_k$ is determined by the following condition,

$$\{2\cos\theta_k + i(l+r)\}\sin N\theta_k - (1+lr)\sin(N-1)\theta_k = 0\,, \tag{24}$$

where we defined $l = \frac{\varepsilon_{\mathrm{L}}}{4J}$ and $r = \frac{\varepsilon_{\mathrm{R}}}{4J}$.

Distribution of the solutions to (24), and hence that of the eigenvalues of the matrix $\Xi$, depend strongly on boundary dissipative strength $\varepsilon_{\mathrm{L/R}}$. When $\varepsilon_{\mathrm{L}} = \varepsilon_{\mathrm{R}} = 0$, the solution of (24) is simply given by $\theta_k = \pi k/(N+1)$, $0 \leq k \leq N$ and the corresponding eigenvalues (22) are distributed on the real axis from $B - 2J$ to $B + 2J$. On the other hand, when $\varepsilon_{\mathrm{L/R}}$ are non-zero, the solutions to (24) and hence the corresponding eigenvalues (22) become complex. In particular when $\varepsilon_{\mathrm{L/R}}$ are larger than $4J$, while most eigenvalues are still close to the real axis, there appear special eigenvalues which have larger imaginary part than the other ones, as shown in an example in Fig. 1. Behaviors of eigenvalues in the limit $N \to \infty$ may be discussed as follows. First using the knowledge of the recurrence relation for the matrix $\Xi$, we obtain the characteristic equation of the matrix $\Xi$ as

$$\beta^{N+1} - \alpha^{N+1} + i(l+r)(\beta^N - \alpha^N) - lr(\beta^{N-1} - \alpha^{N-1}) = 0\,, \tag{25}$$

where $\alpha + \beta = \frac{\lambda - B}{J}$ and $\alpha\beta = 1$. Therefore, if we can solve the equation (25), we obtain the eigenvalue $\lambda = B + J(\beta + \beta^{-1})$. As discussed in [29], the solutions of the above equation when $N \to \infty$ depend on the magnitude of $\beta$. When $|\beta| > 1$, terms containing $\alpha^N$ become small since $|\alpha| < 1$ and the equation (25) becomes

$$\beta^2 + i(l+r)\beta - lr = 0\,. \tag{26}$$

This can be solved easily and the solutions are given by $\beta = -il, -ir$. Hence these solutions exist only when $\varepsilon_{\mathrm{L/R}} > 4J$. In a similar manner, when $|\beta| < 1$, the solutions of equation (25) are given by $\beta = il^{-1}, ir^{-1}$. Lastly, when $|\beta| = 1$, the solution of equation (25) is in the form $\beta = e^{i\theta}$, $\theta \in \mathbb{R}$. We call the eigenvalues without imaginary part $\mathrm{Im}(\lambda) = 0$ normal eigenvalue expressed as $\lambda = B + 2J\cos\theta$ and the eigenvalues with imaginary part $\mathrm{Im}(\lambda) \neq 0$ special eigenvalue expressed as

$$\lambda = B - iJ\left(l - l^{-1}\right),\ B - iJ\left(r - r^{-1}\right). \tag{27}$$

For a large but finite $N$, there appear eigenvalues close to the real axis and the ones with larger imaginary part. The former is expected to become normal eigenvalues and the latter special eigenvalues as $N \to \infty$. They will be called the normal and special eigenvalues respectively even when $N$ is large but not infinite.

Since the matrix $\Xi$ is a complex symmetric matrix, we can diagonalize it by using a complex orthogonal matrix $\mathbf{Q}$ as

$$\Xi = \mathbf{QDQ}^{\mathrm{T}}, \tag{28}$$

where

$$\mathbf{D} = \mathrm{diag}[\lambda^{(1)}, \cdots, \lambda^{(N)}], \ \mathbf{Q} = \left[\underline{Q}^{(1)}, \cdots, \underline{Q}^{(N)}\right]. \tag{29}$$

Denoting the normalization factor of the $k-$th eigenvector by $\mathcal{N}_k \equiv \underline{q}^{(k)} \cdot \underline{q}^{(k)}$, we set $\underline{Q}^{(k)} = \frac{\underline{q}^{(k)}}{\mathcal{N}_k}$. Then, by Lemma 1 and the diagonalization above, the matrix $\mathbf{X}$ is digonalizable as follows,

$$\mathbf{X} = \mathbf{S}^{\dagger}\left(\begin{array}{cc} \mathbf{Q} & 0 \\ 0 & \overline{\mathbf{Q}} \end{array}\right)\left(\begin{array}{cc} i\mathbf{D} & 0 \\ 0 & -i\mathbf{D}^{\dagger} \end{array}\right)\left(\begin{array}{cc} \mathbf{Q}^{\mathrm{T}} & 0 \\ 0 & \mathbf{Q}^{\dagger} \end{array}\right)\mathbf{S}. \tag{30}$$

Also, the matrix $\mathbf{X}$ can be rewritten in a Jordan canonical form,

$$\mathbf{X} = \mathbf{P}\Delta\mathbf{P}^{-1}, \tag{31}$$

where $\mathbf{P}$ is a non-singular matrix, and $\Delta$ is a Jordan canonical form. Let any Jordan cell size be bigger than 1, and the component of the matrix $\mathbf{P}$ be the generalized eigenvectors of the matrix $\mathbf{X}$. Thus, if and only if the matrix $\mathbf{X}$ is diagonalizable, we can consider that these representation are the same. Then, by using (30,31), we obtain the non-singular matrix $\mathbf{P}$ and its inverse matrix $\mathbf{P}^{-1}$ as the follows,

$$\mathbf{P} = \mathbf{S}^{\dagger}\left(\begin{array}{cc} \mathbf{Q} & 0 \\ 0 & \overline{\mathbf{Q}} \end{array}\right), \quad \mathbf{P}^{-1} = \left(\begin{array}{cc} \mathbf{Q}^{\mathrm{T}} & 0 \\ 0 & \mathbf{Q}^{\dagger} \end{array}\right)\mathbf{S}. \tag{32}$$

To summarize the results of this section, we have determined the exact formula of the eigenvalues of the Lindbradian $\mathcal{L}$ in (2). More precisely, we wrote the Lindblad map $\hat{\mathcal{L}}$ in (12) acting on the operator space in the form (13) with (15) and (16), and have obtained the eigenvalues and the corresponding eigenvectors of the matrix $\Xi$ as in (22) and (23) with (24).

## 3 Exact solutions for time-dependence of physical observables

In this section we calculate the exact formulas of time-dependent physical observables by using the exact formula of the eigenvalues and the corresponding eigenvector of the matrix $\Xi$ in the previous section. In this paper we focus on the time-dependent magnetization and spin current. The results will be summarized as (45,46) below,

### 3.1 Exact formulas for magnetization and current

The physical observables $X(t)$ at time $t$ is defined in Schrödinger picture as $X(t) = \mathrm{tr}(X\rho(t))$ [51,53,54]. Since the Lindbladian in Hilbert space is difficult to study analytically, we consider the Heisenberg picture in Liouville-Fock space [70,71]. As presented below in (36,37), the time-dependent magnetization and spin current are written in terms of quadratic physical observables [51,54] defined as

$$\mathsf{C}_{j,k}(t) = \mathrm{tr}\big(w_j w_k \rho(t)\big). \tag{33}$$

Then, since the diagonal terms in $C_{j,k}(t)$ are time-invariant $C_{j,j}(t) = \text{tr}(\rho(t)) = \text{tr}(\rho(0))$, we define the correlation matrix $\tilde{\mathbf{C}}(t) = \left\{ \tilde{C}_{j,k}(t) \right\}_{1 \leq j,k \leq N}$ by

$$\tilde{C}_{j,k}(t) = \text{tr}\left( w_j w_k \rho(t) \right) - \delta_{j,k} = 2 \langle 1 | \hat{a}_{1,j}(t) \hat{a}_{1,k}(t) | \rho_0 \rangle - \delta_{j,k}, \tag{34}$$

where the super-Heisenberg picture is defined by $\hat{a}_k(t) := e^{-t\hat{\mathcal{L}}} \hat{a}_k e^{t\hat{\mathcal{L}}}$. Using the Lindbladian map $\hat{\mathcal{L}}$ (13), we can obtain the equation of motion for Majorana map as follows,

$$\frac{d\underline{\hat{a}}(t)}{dt} = 2\mathbf{A}\underline{\hat{a}}(t). \tag{35}$$

In terms of $\tilde{C}_{j,k}(t)$, the magnetization $m_k^z(t)$ on site $k$ and the spin current $j_{k,k+1}(t)$ between sites $k$ and $k+1$ can be written by using (5) as follows,

$$m_k^z(t) = \left\langle \sigma_k^z \right\rangle(t) = -i\tilde{C}_{2k-1,2k}(t), \tag{36}$$

$$j_{k,k+1}(t) = \left\langle 2J(\sigma_k^x \sigma_{k+1}^y - \sigma_k^y \sigma_{k+1}^x) \right\rangle(t) = -2Ji\tilde{C}_{2k-1,2k+1}(t) - 2Ji\tilde{C}_{2k,2k+2}(t). \tag{37}$$

The time-dependent correlation matrix $\tilde{\mathbf{C}}(t)$ satisfies the following differential equation [70, 71],

$$\frac{d\tilde{\mathbf{C}}(t)}{dt} = -2\left\{ \mathbf{X}^\mathsf{T}\tilde{\mathbf{C}}(t) + \tilde{\mathbf{C}}(t)\mathbf{X} \right\} - 4i\mathbf{M}_I. \tag{38}$$

Since the components of the matrix $\tilde{\mathbf{C}}(t)$ correspond to the physical observables as (36,37), obtaining the exact solution $\tilde{\mathbf{C}}(t)$ (34) is equivalent to obtaining the exact formulas of the time-dependent the physical observables. In some papers [70–73], this equation (38) has been solved numerically or only its steady state ($\frac{d\tilde{\mathbf{C}}(t)}{dt} = 0$) has been examined, since exact eigenvalues and eigenvectors for the open XX spin chain have not been obtained. By using the exact spectrum of the matrix $\Xi$ (20) and the solvability of this equation (38) [58–60] which had been known in a different field, such as the control theory [74, 75] and stability analysis [76], we can solve this equation and obtain the time-dependence of the physical observables analytically for the first time.

As shown in [58–60], the time-dependence of the correlation matrix is

$$\tilde{\mathbf{C}}(t) = e^{-2t\mathbf{X}^\mathsf{T}}\tilde{\mathbf{C}}(0)e^{-2t\mathbf{X}} + \int_0^t e^{-2s\mathbf{X}^\mathsf{T}}(-4i\mathbf{M}_I)e^{-2s\mathbf{X}}\,ds. \tag{39}$$

For the above formula (39), we can calculate the exact solution for the time-dependence of the correlation matrix $\tilde{\mathbf{C}}(t)$, if the eigenvalues and the (general) eigenvectors of the matrix $\mathbf{X}$ can be exactly calculated and the correlation matrix in the initial time $\tilde{\mathbf{C}}(0)$ can be determined analytically. For the open XX spin chain, we can obtain the eigenvalues and the (general) eigenvectors of the matrix $\mathbf{X}$ can be exactly calculated. Thus, when we choose the correlation matrix in the initial time $\tilde{\mathbf{C}}(0)$ whose components can be determined analytically, we can obtain the exact solution for the time-dependence of the physical observables, and discuss their behaviors.

In this paper, we introduce the time-dependence from one of the simplest initial states satisfying the condition about the correlation matrix in the initial time $\tilde{\mathbf{C}}(0)$. We choose the thermal equilibrium state in the high-temperature limit ($\beta \to 0$) as the initial state. Then, the correlation matrix $\tilde{\mathbf{C}}(t)$ in (34) at the time $t = 0$ becomes zero $\tilde{C}_{j,k}(0) = 0$. Thus, the time-dependence of the correlation matrix takes the following form,

$$\tilde{\mathbf{C}}(t) = \int_0^t e^{-2s\mathbf{X}^\mathsf{T}}(-4i\mathbf{M}_I)e^{-2s\mathbf{X}}\,ds. \tag{40}$$

For the open XX spin chain, since the matrix $\mathbf{X}$ is diagonalizable $\mathbf{X} = \mathbf{P}\Delta\mathbf{P}^{-1}$, the correlation matrix is calculated as

$$\tilde{\mathbf{C}}(t) = \mathbf{P}^{-\mathrm{T}}\left(\left(\int_0^t e^{-2s(\beta_i+\beta_j)}\,\mathrm{d}s\right)_{i,j=1,\cdots,2N} \odot \left(\mathbf{P}^{\mathrm{T}}(-4i\mathbf{M}_I)\mathbf{P}\right)\right)\mathbf{P}^{-1}, \tag{41}$$

where $\beta_j$ is an eigenvalue of the matrix $\mathbf{X}$, and we define the Hadamard product as $(\mathbf{A}\odot\mathbf{B})_{i,j} = A_{i,j}B_{i,j}$. Moreover, since the eigenvalues of $\mathbf{X}$ are calculated from the eigenvalues of the matrix $\Xi$ from the Corollary 1 and the imaginary parts of the eigenvalues of the matrix $\Xi$ are negative, the real parts of the eigenvalues of the matrix $\mathbf{X}$ are positive $\mathrm{Re}\{\beta_j\} > 0$. Thus, the integral in (41) can be calculated as

$$\int_0^t e^{-2s(\beta_i+\beta_j)}\,\mathrm{d}s = \frac{1-e^{-2t(\beta_i+\beta_j)}}{2(\beta_i+\beta_j)}. \tag{42}$$

Therefore, we obtain

$$\tilde{\mathbf{C}}(t) = \mathbf{P}^{-\mathrm{T}}\left(\left(\frac{1-e^{-2t(\beta_i+\beta_j)}}{2(\beta_i+\beta_j)}\right)_{i,j=1,\cdots,2N} \odot \left(\mathbf{P}^{\mathrm{T}}(-4i\mathbf{M}_I)\mathbf{P}\right)\right)\mathbf{P}^{-1}. \tag{43}$$

The magnetization $m_k^z(t)$ takes the following form,

$$m_k^z(t) = \sum_{n,m=1}^{2N}\frac{e^{-2t(\beta_m+\beta_n)}-1}{2(\beta_m+\beta_n)}P_{2k-1,m}^{-\mathrm{T}}\left[\mathbf{P}^{\mathrm{T}}(4\mathbf{M}_I)\mathbf{P}\right]_{m,n}P_{n,2k}^{-1}. \tag{44}$$

Substituting imaginary part of dissipative matrix $\mathbf{M}$, non-singular matrix $\mathbf{P}$ and that inverse matrix $\mathbf{P}^{-1}$ (32) to (44), the magnetization in (36) takes the following spectral decomposition form,

$$m_k^z(t) = \sum_{m,n=1}^{N}\mathrm{Re}\left[\frac{1-e^{-2it(\lambda^{(m)}-\lambda^{(n)*})}}{2i(\lambda^{(m)}-\lambda^{(n)*})}Q_k^{(m)}\left\{\varepsilon_\mathrm{L}\mu_\mathrm{L}Q_1^{(m)}Q_1^{(n)*}+\varepsilon_\mathrm{R}\mu_\mathrm{R}Q_N^{(m)}Q_N^{(n)*}\right\}Q_k^{(n)*}\right]. \tag{45}$$

Similarly, spin current between sites $k$ and $k+1$ $j_{k,k+1}(t)$ in (37), and takes the following spectral decomposition form,

$$j_{k,k+1}(t) = 4J\sum_{m,n=1}^{N}\mathrm{Im}\left[\frac{1-e^{-2it(\lambda^{(m)}-\lambda^{(n)*})}}{2i(\lambda^{(m)}-\lambda^{(n)*})}Q_k^{(m)}\left\{\varepsilon_\mathrm{L}\mu_\mathrm{L}Q_1^{(m)}Q_1^{(n)*}+\varepsilon_\mathrm{R}\mu_\mathrm{R}Q_N^{(m)}Q_N^{(n)*}\right\}Q_{k+1}^{(n)*}\right]. \tag{46}$$

In (45,46), the eigenvalues $\lambda^{(m)} = -i\beta_m$ and the matrix elements $Q_k^{(m)}$ which is the $k$-th component of the eigenvector corresponding to the eigenvalue $\lambda^{(m)}$ are defined by using (22-24,29) and the definition of the normalization factor $\mathcal{N}_m$ as

$$\lambda^{(m)} = B + 2J\cos\theta_m, \quad Q_k^{(m)} = \frac{\dfrac{1}{\sin\theta_m}\left[\sin k\theta_m + i\dfrac{\varepsilon_\mathrm{L}}{4J}\sin((k-1)\theta_m)\right]}{\sqrt{\displaystyle\sum_{k=1}^{N}\left(\dfrac{1}{\sin\theta_m}\left[\sin k\theta_m + i\dfrac{\varepsilon_\mathrm{L}}{4J}\sin((k-1)\theta_m)\right]\right)^2}}, \tag{47}$$

where $\theta_m$ satisfies the following equation,

$$\left\{2\cos\theta_m + i\left(\frac{\varepsilon_\mathrm{L}}{4J}+\frac{\varepsilon_\mathrm{R}}{4J}\right)\right\}\sin N\theta_m - \left(1+\frac{\varepsilon_\mathrm{L}}{4J}\frac{\varepsilon_\mathrm{R}}{4J}\right)\sin(N-1)\theta_m = 0. \tag{48}$$

The exact formulas (45,46) with (47,48) for time-dependent magnetization in (36) and spin current in (37) are the main results in this paper.

## 3.2 Physical observables in steady state

Before going to discussions of dynamical behaviors, in this subsection, we consider briefly the physical observables in steady state which is realized in the long time limit. Taking the limit $t \to \infty$ in (45,46), magnetization and spin current in steady state are expressed as

$$m_k^z = \sum_{m,n=1}^{N} \mathrm{Re}\left[ \frac{Q_k^{(m)}\left\{\varepsilon_L \mu_L Q_1^{(m)} Q_1^{(n)*} + \varepsilon_R \mu_R Q_N^{(m)} Q_N^{(n)*}\right\} Q_k^{(n)*}}{2i(\lambda^{(m)} - \lambda^{(n)*})} \right], \tag{49}$$

$$j_{k,k+1} = 4J \sum_{m,n=1}^{N} \mathrm{Im}\left[ \frac{Q_k^{(m)}\left\{\varepsilon_L \mu_L Q_1^{(m)} Q_1^{(n)*} + \varepsilon_R \mu_R Q_N^{(m)} Q_N^{(n)*}\right\} Q_{k+1}^{(n)*}}{2i(\lambda^{(m)} - \lambda^{(n)*})} \right], \tag{50}$$

where $\lambda^{(m)}$ and $Q_k^{(m)}$ are given by (47,48). After some calculations, we arrive at the following simple formulas for the magnetization and the spin current for steady state (The detailed calculations are written in Appendix A.) in terms of model parameters (recall $l, r$ defined below (24)),

$$m_k^z = \mu_L - \frac{j}{4J} D_k^{(L)} = \mu_R + \frac{j}{4J} D_k^{(R)}, \quad j = \frac{\varepsilon_L \varepsilon_R (\mu_L - \mu_R)}{4J\left(1 + \frac{\varepsilon_L}{4J}\frac{\varepsilon_R}{4J}\right)\left(\frac{\varepsilon_L}{4J} + \frac{\varepsilon_R}{4J}\right)}, \tag{51}$$

where $D_k^{(L)}$ and $D_k^{(R)}$ are defined as

$$D_1^{(L)} = \frac{4J}{\varepsilon_L}, \quad D_k^{(L)} = \frac{\varepsilon_L}{4J} + \frac{4J}{\varepsilon_L}, \quad (2 \le k \le N-1), \quad D_N^{(L)} = \frac{\varepsilon_L}{4J} + \frac{4J}{\varepsilon_L} + \frac{\varepsilon_R}{4J}, \tag{52}$$

$$D_1^{(R)} = \frac{\varepsilon_R}{4J} + \frac{4J}{\varepsilon_R} + \frac{\varepsilon_L}{4J}, \quad D_k^{(R)} = \frac{\varepsilon_R}{4J} + \frac{4J}{\varepsilon_R}, \quad (2 \le k \le N-1), \quad D_N^{(R)} = \frac{4J}{\varepsilon_R}. \tag{53}$$

Our formulas for the magnetization and the spin current are valid for all parametric values of $\mu_{L/R}$, $\varepsilon_{L/R}$, and agree with the results in [34–36] obtained by MPA for the case of the anti-symmetric magnetization on reservoirs ($\mu_L = -\mu_R$). For Fig.2, we confirm that our formula (51-53) for magnetization and spin current for steady state coincide with the ones obtained by MPA [34–36] (when $\mu_L = -\mu_R$). Our formulas (51-53) are the most general solution for the magnetization and the spin current in steady state for the open XX spin chain with boundary dissipation in the sense that they are valid for all parametric values of $\mu_{L/R}$, $\varepsilon_{L/R}$. It is also interesting to consider whether our results for the general parameters case can also be realized in terms of MPA.

# 4 The dynamics of physical observables

Analytical studies of open quantum systems with Lindblad dynamics for large systems have been challenging, because explicit diagonalization of a Lindbladian is in general difficult, and most studies so far are numerical. For the open XX spin chain with boundary dissipation, the solutions in steady state are obtained by using MPA [34–36], but the dynamics have been much less understood analytically. Since we could diagonalize the Lindbladian in section 2 and obtained the analytical formulas for the time-dependence of magnetization (45) and spin current (46) in section 3, we can study their behaviors in detail.

## 4.1 Behaviors of time-dependent physical observables

We first evaluate our formulas (45,46) numerically and observe several behaviors for the time-dependence of the magnetization and the spin current for the open XX chain. We will examine

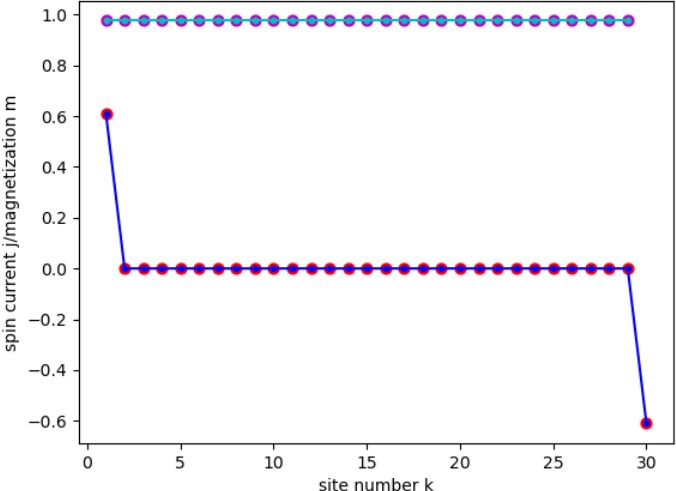

Figure 2: Magnetization (red dots and blue line) and spin current for steady state (magenta dots and cyan lines). The red and magenta dots are obtained by our formula (51-53), and the blue and cyan lines are obtained in [34–36], respectively. The parameters are set to $N = 30$, $J = 1.0$, $B = 0.0$, $\varepsilon_{\mathrm{L/R}} = 5$ and $\mu_{\mathrm{L}} = -\mu_{\mathrm{R}} = 1$.

them analytically in subsequent discussions. In Fig.3, spatio-temporal behaviors of the magnetization are displayed. We observe a clear and interesting light-cone structure. In Fig.4, the time-dependence of the magnetization and the spin current are plotted for several fixed sites with label $k$. Behaviors of the physical quantities depend on the position of a site $k$ in the system. At the beginning, at sites near a boundary, the magnetization and the spin current show a rather clear plateau regime as in Figs.4a and 4c. It appears as soon as the time evolution starts and the magnetization almost does not change during it. It also appears at a bulk site but becomes shorter and obscure near the center of the chain. See Figs.4b and 4d. After the plateau regime, the physical quantities show a few steps of small plateaus with oscillations and then decay to the stationary values.

About shorter time behaviors, a basic mechanism of the appearance of the plateau regime at a given site may be understood from the wave fronts of the light-cone structures in Fig.3. From the initial time, effects of the boundary dissipations propagate along the bulk part of the system, creating the light-cones. The slope of the light cones is expected to be given by $1/4J$, which is numerically checked, and may be interpreted as the fastest group velocity within all the group velocities for this system as will be discussed in section 4.2.1. According to this picture, the plateau regime becomes shorter and shorter as a site deviates from a boundary and vanishes at the site at the center of the chain. These behaviors are seen in the short time region ($0 \leq 4Jt \lesssim \mathcal{O}(N^1)$).

Approach to stationary values of the physical observables after a very long time ($t \gg 1$) is expected to be described by the Liouvillian gap, which is the spectral gap $\Delta$ of the Liouvillian [51,54]. The finite-size scaling for the Liouvillian gap had been numerically estimated [51–53, 77], and we examine it analytically by using the exact formula of the eigenvalues of the matrix $\Xi$ (20) for our system. In the time region after the plateau regime and before the Liouvillian gap dominates the decay of physical quantities, ($\mathcal{O}(N^1) \lesssim 4Jt \lesssim \mathcal{O}(N^3)$), physical quantities show rather complicated behaviors.

In the following, we analytically discuss these behaviors by using our formulas (45,46). We first discuss the two short-time behaviors ($0 \leq 4Jt \lesssim \mathcal{O}(N^1)$). The one is the light-cone structures in Fig.3 using the analogy to that in closed systems. The other is the plateau regime. Second, we analytically estimate the finite-size scaling for the Liouvillian gap.

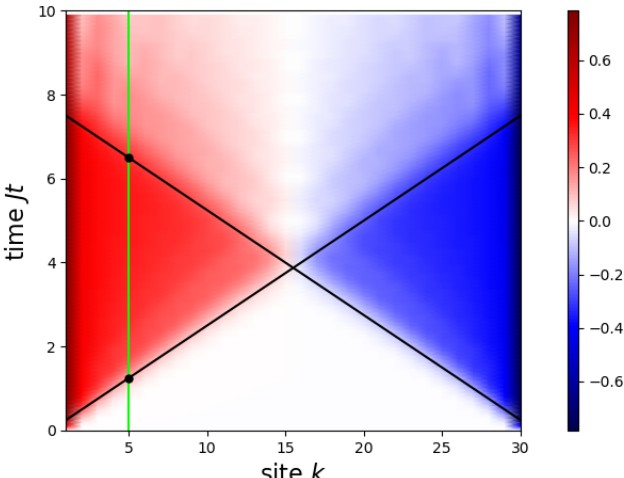

Figure 3: Spatio-temporal dependence of the local magnetization by (45). The parameters are set to $N = 30$, $\varepsilon_{L/R} = 5.0$, $J = 1.0$, $B = 0.0$, $\mu_L = -\mu_R = 1.0$. The black lines are $Jt = k/4$ and $Jt = (N - k + 1)/4$ which represent the initial and final time of the plateau regime. The points at the intersection of the green and black lines are the initial and final time of the plateau regime at the site 5. We show details of the analysis of the plateau regime later in this section.

## 4.2 The short time behaviors ($0 \sim 4Jt \sim \mathcal{O}(N^1)$)

### 4.2.1 Light-cone structure

For quench dynamics of various quantum many-body systems, it has been discovered that frontiers of local observables show a light-cone structure whose slope should be bounded above by the Lieb-Robinson velocity [11, 63, 78, 79]. In particular, for the quench dynamics in the closed XX spin chain, the light-cone appears from the free magnon propagation. Its propagating velocity is calculated as the group velocity $|v| = |d\varepsilon(k)/dk|$ from the dispersion relation $\varepsilon = \varepsilon(k) = J\cos k$ of the one particle excitation [64, 79], where $k$ is a momentum and $\varepsilon$ is an eigenenergy, and the slope of the light-cone is given by its maximum, taken at $k = \pi/2$.

The slope of the light-cones in Fig.3 for our open XX spin chain may be determined by using an analogy to the quench dynamics in the closed XX spin chain discussed above. More precisely we may conjecture that eigenvalue $\lambda^{(m)}$ of $\Xi$ (20,22) would play a similar role as eigenenergy $\varepsilon(k)$ and that the propagation speed of the $m$-th mode is given by the formula,

$$|v| = \left| \frac{d(2\lambda^{(m)})}{d\theta_m} \right| = |4J\sin\theta_m|, \tag{54}$$

where $\theta_m$ is determined as (24). This is plausible because the dependence of physical quantities such as the magnetization on the eigenvalue $\lambda^{(m)}$ of $\Xi$ (20,22), given in (45,46), for our case of Lindblad dynamics is similar to the one on $\varepsilon$ for the case of quench dynamics. The factor 2 in front of $\lambda_m$ in (54) may be attributed to the same factor in (35), which could be absorbed in the definition of the Majorana fermion operator by changing the inner product which the operator space $\mathcal{K}$ is orthonormal with respect to [51, 54]. From discussions about distributions of $\theta_k$ around (22), the velocity approaches, at $\theta_m \approx \pi/2$, the maximum value $|v|_{max} = 4J$, and the fastest propagation of all effects of each boundary dissipation has this velocity. Thus the slope of the sharp front in Fig.3 is supposed to be a quarter with the dimensionless time unit $Jt$ in Fig.3, and this is numerically indeed confirmed. Moreover, by carefully examining our formula (45), we will derive the slope of the light-cone in the next subsection.

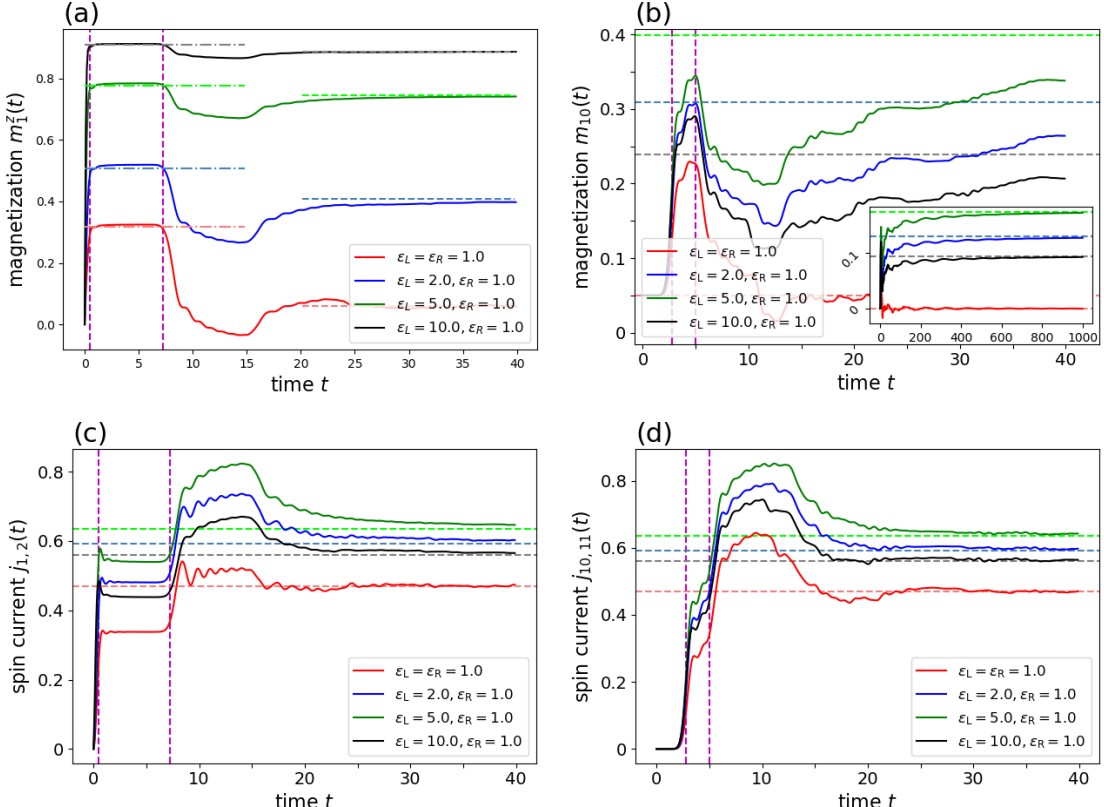

Figure 4: Time-dependence of the magnetization ((a) and (b)) given by (45) and the current ((c) and (d)) given by (46) respectively. Different colors in each figure correspond to different values of dissipative strength $\varepsilon_{\mathrm{L}}$, i.e., the red, blue, green and black curves correspond to $\varepsilon_{\mathrm{L}} = 1.0, 2.0, 5.0, 10.0, \varepsilon_{\mathrm{R}} = 1.0$ cases, respectively. (a) and (c): The time-dependence of the magnetization and the spin current near the left boundary ($k = 1$). (b) and (d): The time-dependence of the magnetization and the spin current at a bulk site ($k = 10$). The inset in (b) is for a longer time scale. The magenta vertical lines describe start and finish time of the plateau regime which is expected from the light-cone structures. The light color dashed lines are the magnetization and spin current in the steady state. The light color dash-dotted lines in (a) show the plateau heights calculated by (58). Other parameters in these pictures are set to $N = 30, J = 1.0, B = 0.0, \mu_{\mathrm{L}} = -\mu_{\mathrm{R}} = 1.0$.

### 4.2.2 The emergence of the plateau regime

For understanding behaviors of the plateau regime, during which the magnetization does not change, we calculate the time derivative of magnetization for the site $k$ from (45) as

$$\mu_k(t) := \frac{\partial m_k}{\partial t} = \varepsilon_{\mathrm{L}} \mu_{\mathrm{L}} \left| \sum_{n=1}^{N} e^{-2it\lambda^{(n)}} Q_1^{(n)} Q_k^{(n)} \right|^2 + \varepsilon_{\mathrm{R}} \mu_{\mathrm{R}} \left| \sum_{n=1}^{N} e^{-2it\lambda^{(n)}} Q_N^{(n)} Q_k^{(n)} \right|^2, \quad (55)$$

where label $n$ represents the mode number. The first and the second terms in (55) will be called the left and right dissipation contributions, respectively. For large system $N \gg 1$, after some calculations (see Appendix. B for details), we obtain

$$\sum_{n=1}^{N} e^{-2it\lambda^{(n)}} Q_j^{(n)} Q_k^{(n)} \approx f(j,k;t) := \oint_C \frac{\mathrm{d}z}{2\pi i} e^{2Jt(z-z^{-1})} \left\{ \frac{i^{k-j}}{z^{k-j-1}} + \frac{i^{j+k}(z+l)z^{j+k-2}}{lz-1} \right\}, \quad (56)$$

the parameter $l$ is defined below (24) and the contour $C$ is such that it encloses the origin counter clockwise with radius less than $1/l$. The function $f(j, k; t)$ may also be written as a series in terms of the Bessel functions (see (B.8)), but the contour integral expression is more convenient for our discussions below. For large $N$, (55) is approximated in terms of $f(j, k; t)$ as

$$\mu_k(t) \approx \varepsilon_{\mathrm{L}} \mu_{\mathrm{L}} |f(1, k; t)|^2 + \varepsilon_{\mathrm{R}} \mu_{\mathrm{R}} |f(N, k; t)|^2 . \tag{57}$$

Now let us focus on $f(1, k; t)$. As we show in Appendix B, it is close to zero except near $t \sim k/(4J)$. By the same reasoning, the function $f(N, k; t)$ is close to zero except near $x \sim (N - k)/(4J)$. This confirms the slope $1/4J$ of the light cone, mentioned at the end of section 4.2.1.

From the above analysis we see that, between $t = \frac{k}{4J}$ and $t = \frac{N-k+1}{4J}$, the time derivative of the magnetization $\mu_k(t)$ is almost equal to zero, i.e., the magnetization does not change. Then the duration of time between $t = \frac{k}{4J}$ and $t = \frac{N-k+1}{4J}$ may be identified as the plateau region. Its duration time $\tau_p = \left| \frac{N-2k-1}{4J} \right|$ decreases to zero as the site becomes closer to the center of the system. This prediction of the plateau regions agree well with the figures of physical quantities (see Fig. 4).

While the clear plateau is seen near the boundaries (see for instance Fig .4a), additional smaller changes are observed on top of the plateau in the bulk (see for instance Fig 4b). This may be explained by the fact that the period of oscillatory behaviors of $f(1, k; t)$ become small when $k$ is large, see (B.17).

The height of the plateau regime can also be calculated by using the time-derivative of the magnetization $\mu_k(t)$ (57). Since the height of the plateau regime depends on either the left or right contribution to the time derivative of the magnetization $\mu_k(t)$ (57), we only consider the left half of the system. In this case, the height of the plateau regime $H_p(J, \varepsilon_{\mathrm{L}}, \mu_{\mathrm{L}}; k)$ at the site $k$ in the small dissipative case ($\varepsilon_{\mathrm{L}} < 4J$) is estimated as

$$
H_p(J, \varepsilon_{\mathrm{L}}, \mu_{\mathrm{L}}; k) \approx \frac{4\mu_{\mathrm{L}}}{1 - l^{-2}} + 2\mu_{\mathrm{L}} \sum_{p,q,m,n=0}^{\infty} (-l)^{p+1} (-1)^{n+m}
$$
$$
\times \left( J_{k+n+p-q-1}(T_i^{(k)}) + J_{k+n+p-q+1}(T_i^{(k)}) \right) \left( J_{k+m-p-q-1}(T_i^{(k)}) + J_{k+m-p-q+1}(T_i^{(k)}) \right),
\tag{58}
$$

where $T_i^{(k)} = 4J\tau_i^{(k)} \approx k + 1$ and $\tau_i^{(k)}$ is an initial time for the plateau regime at the site $k$. For obtaining this formula, we use the integral form of the Bessel function. Of course, other cases, such as $\varepsilon_{\mathrm{L}} \geq 4J$, can be derived by using a similar procedure. The formula is almost exact numerically (see Fig. 4a), with a small error due to finite-size effects. A physical interpretation of the formula for the height is not very clear for the moment.

## 4.3 Long time behaviors ($4Jt \sim \mathcal{O}(N^3)$) and Liouvillian gap

For systems described by the Lindblad equation, asymptotic long time behaviors of physical observables are in general expected to be characterized by the Liouvillian gap [77]. See for instance [51–53, 80]. The double of the Liouvillian gap, denoted by $\Delta$, is for our system defined as

$$\Delta = -2 \max \mathrm{Im}[\lambda^{(n)}], \tag{59}$$

where $\lambda^{(n)}$ is the eigenvalue of the matrix $\Xi$ which is defined as (22,24). The relaxation time $\tau$ of the system is determined as the inverse of $\Delta$. The slow convergence at late times, observed in Fig.4b, is expected to have this relaxation time. In Fig.5, we show more precise

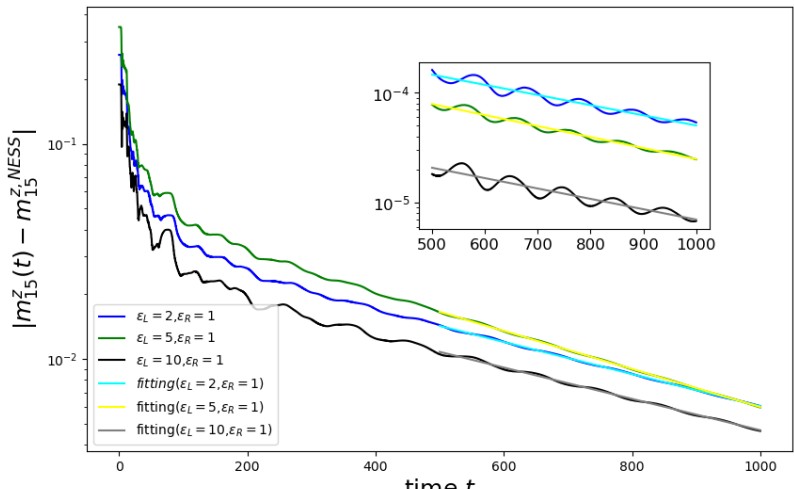

Figure 5: Asymptotic behavior of the magnetization after long time: The time-dependence of the magnetization at the site $k = 15$ when $N = 30, J = 1.0, B = 0.0, \mu_L = \mu_R = 1.0$. The cyan, yellow and gray lines represent the fittings to exponential decays. The values of their slopes are given in the text. The inserted figure exhibits the time-dependence of the magnetization at the site 1 and the exponential functions fitted to data.

semi-logarithmic plots of the difference of the magnetization at time $t$ to its steady state value. We observe indeed that its asymptotic behaviors at a site $k$ becomes an exponential decay. In our numerical results, the inverse of the relaxation times $1/\tau$ which are obtained by fitting to the data for the time-dependence of the magnetization from $t = 500$ to $t = 1000$ using our formula (45) are $1.71_7 \times 10^{-3} (\varepsilon_L = 2.0, \varepsilon_R = 1.0)$, $2.03_5 \times 10^{-3} (\varepsilon_L = 5.0, \varepsilon_R = 1.0)$, and $1.67_5 \times 10^{-3} (\varepsilon_L = 2.0, \varepsilon_R = 1.0)$, which should be compared to twice the Liouvillian gaps $2\Delta = 1.72_1 \times 10^{-3} (\varepsilon_L = 2.0, \varepsilon_R = 1.0)$, $2.04_1 \times 10^{-3} (\varepsilon_L = 10.0, \varepsilon_R = 1.0)$, and $1.67_8 \times 10^{-3} (\varepsilon_L = 10.0, \varepsilon_R = 1.0)$ computed using (59). We see that the relaxation time $\tau$ for these exponential decay agree well with the inverse of the double of the Liouvillian gap $\tau \approx 1/2\Delta$.

In previous studies, the Liouvillian gaps for open quantum systems have not been much discussed analytically. They have been mostly calculated numerically or by using the analogy from closed infinite systems. For example the Liouvillian gap for open transverse Ising spin chain and XY spin chain has been estimated by the asymptotic result using analogy from closed infinite systems [51, 52, 81]. For our case of the open XX spin chain, we can estimate the magnitude of the Liouvillian gap $\Delta$ by using the exact spectrum of the Lindbladian for a finite-size system obtained in section 2.2. The Liouvillian gap corresponds to $n = 1$ case of the eigenvalue $\lambda^{(n)}$, defined as (22,24). Let us write the angle $\theta_1$ in (24) as $\theta_1 = \frac{\pi}{N+1} + \frac{x+iy}{N^2}$. Substituting this into (24) and keeping the leading order terms in $1/N$, we see that $x$ and $y$ are given by

$$x = \frac{l^2 + 2l^2 r^2 + r^2}{(l+r)^2 + (lr-1)^2} \pi, \quad y = \frac{(l+r)(lr+1)}{(l+r)^2 + (lr-1)^2} \pi. \quad (60)$$

Recalling the definitions of $l, r$ which were given below (24), the Liouvillian gap can be expressed in terms of model parameters as follows,

$$\Delta = 4J\pi^2 \frac{\left(\frac{\varepsilon_L}{4J} + \frac{\varepsilon_R}{4J}\right)\left(\frac{\varepsilon_L}{4J}\frac{\varepsilon_R}{4J} + 1\right)}{\left(\frac{\varepsilon_L}{4J} + \frac{\varepsilon_R}{4J}\right)^2 + \left(\frac{\varepsilon_L}{4J}\frac{\varepsilon_R}{4J} - 1\right)^2} \frac{1}{N^3}. \quad (61)$$

The $O(N^{-3})$ behaviors have been observed numerically in [48, 51, 52, 81] but we have con-

firmed it and have also determined the coefficient exactly. Our formula shows great agreement with numerical diagonalization, including the coefficient.

# 5 Conclusion

We have applied the general procedure of the third quantization to the open XX spin chain. We find that the structure matrix of the open XX spin chain is diagonalizable analytically. Moreover, we find that although the structure matrix is ordinarily decomposed into $2N \times 2N$ matrix, the structure matrix for the open XX spin chain is decomposed into $N \times N$ non-Hermitian matrix, and the eigenvalues and eigenvectors of this non-Hermitian matrix are calculated analytically. The eigenvalue distribution of this non-Hermitian matrix changes dramatically with the increment of boundary dissipative strength. If a dissipative strength is larger than four times the coupling constant between sites on the system, we could find the emergence of a special eigenvalue which has a larger imaginary part than the others. Since the open XX spin chain is diagonalizable, we can exactly calculate time-evolution from a general initial condition including the thermal equilibrium state. We obtain the linear differential equation for the correlation matrix which is constructed from the expectation value of the product of two Majorana operators. The several components of the correlation matrix correspond to magnetization on the site $k$ and spin current between the site $k$ and $k+1$.

The exact solutions of time-dependent magnetization on arbitrary site $k$ and spin current between arbitrary sites $k$ and $k+1$ are the main results of this study. These formulas also include the solutions for NESS which is defined as $t \to \infty$. Our analytical formulas for magnetization and spin current in steady state generalize the ones obtained by the MPA solutions for the special case of antisymmetric magnetization on reservoirs [34–36]. Evaluating the exact solutions of time-dependent magnetization on arbitrary site $k$ and spin current between arbitrary sites $k$ and $k+1$ numerically, we observe some specific behaviors. Using our formulas, we can examine these analytically. As the spatio-temporal regions where the magnetization is large are displayed, we observe clear and interesting light-cone structures. We have shown that the wave fronts of the light-cones for our open XX spin chain can be determined by using an analogy to the quench dynamics in the closed XX spin chain. Between the lightcones from the left and from the right, there appears the plateau regime, during which the magnetization does not change. Its duration is called the plateau time. Various properties of the plateau regime, such as the plateau time, have been clarified by performing the asymptotic analysis of integral formulas for the time derivative of magnetization. After the plateau regime, physical quantities approach their stationary values, with the relaxation time characterized by the Liouvillian gap. We could not only establish its $O(N^{-3})$ behavior, which had been observed, but also determine its coefficient exactly from our formulas.

It is important that one can obtain the exact formula for the time-dependence of physical observables analytically. Applying this fact, higher-order physical observables will be calculated analytically. Moreover, since the Lindbladian map takes the Jordan canonical form in an arbitrary quadratic fermion chain, XY spin chain, XX spin chain with homogeneous bulk dissipation and long-range interaction systems can be analyzed. Recently, the analysis of non-Hermitian systems has been applied to open quantum systems by using the post-selection [82, 83], and many interesting properties for open quantum systems, such as phase transitions [84] and topological natures [85] have been studied. However, it has been known the dynamics which is described by the non-Hermitian systems is different from the Lindblad dynamics [82, 83]. We hope that our exact results will be useful for future studies of systems described by the Lindblad equation. Our studies in this paper are fully based on exact calculations for microscopic models. It would be also interesting to study similar dynamical behaviors of open quantum

systems by using macroscopic or hydrodynamical methods. Some studies in such a direction have recently been performed, see for instance [86–88].

## Acknowledgments

The authors are grateful to T. Fukadai and Y. Nakanishi for useful discussions. The work of TS is supported by JSPS KAKENHI Grants No. JP16H06338, No. JP18H01141, No. JP18H03672, No. JP21H04432, No. JP22H01143.

## A    Physical observables for steady state

In the main part of the paper, we find formulas for magnetization and spin current for steady state as follows,

$$m_k^z = \sum_{m,n=1}^{N} \mathrm{Re}\left[\frac{1}{2i(\lambda^{(m)}-\lambda^{(n)*})}Q_k^{(m)}\left\{\varepsilon_{\mathrm{L}}\mu_{\mathrm{L}}Q_1^{(m)}Q_1^{(n)*}+\varepsilon_{\mathrm{R}}\mu_{\mathrm{R}}Q_N^{(m)}Q_N^{(n)*}\right\}Q_k^{(n)*}\right], \quad (A.1)$$

$$j_{k,k+1} = 4J\sum_{m,n=1}^{N} \mathrm{Im}\left[\frac{1}{2i(\lambda^{(m)}-\lambda^{(n)*})}Q_k^{(m)}\left\{\varepsilon_{\mathrm{L}}\mu_{\mathrm{L}}Q_1^{(m)}Q_1^{(n)*}+\varepsilon_{\mathrm{R}}\mu_{\mathrm{R}}Q_N^{(m)}Q_N^{(n)*}\right\}Q_{k+1}^{(n)*}\right], \quad (A.2)$$

where eigenvector $\beta_j = i\lambda^{(j)}$ and the component of eigenvector $Q_k^{(j)}$ is obtained (22,23), and the parameter $\theta_j$ satisfies the conditional equation (24). Then, separating left and right boundary contributions,

$$m_{k,\mathrm{L}}^z = \varepsilon_{\mathrm{L}}\mu_{\mathrm{L}}\sum_{m,n=1}^{N} \mathrm{Re}\left[\frac{1}{2i(\lambda^{(m)}-\lambda^{(n)*})}Q_k^{(m)}Q_1^{(m)}Q_1^{(n)*}Q_k^{(n)*}\right], \quad (A.3)$$

$$m_{k,\mathrm{R}}^z = \varepsilon_{\mathrm{R}}\mu_{\mathrm{R}}\sum_{m,n=1}^{N} \mathrm{Re}\left[\frac{1}{2i(\lambda^{(m)}-\lambda^{(n)*})}Q_k^{(m)}Q_N^{(m)}Q_N^{(n)*}Q_k^{(n)*}\right], \quad (A.4)$$

$$j_{k,k+1,\mathrm{L}} = 4J\varepsilon_{\mathrm{L}}\mu_{\mathrm{L}}\sum_{m,n=1}^{N} \mathrm{Im}\left[\frac{1}{2i(\lambda^{(m)}-\lambda^{(n)*})}Q_k^{(m)}Q_1^{(m)}Q_1^{(n)*}Q_{k+1}^{(n)*}\right], \quad (A.5)$$

$$j_{k,k+1,\mathrm{R}} = 4J\varepsilon_{\mathrm{R}}\mu_{\mathrm{R}}\sum_{m,n=1}^{N} \mathrm{Im}\left[\frac{1}{2i(\lambda^{(m)}-\lambda^{(n)*})}Q_k^{(m)}Q_N^{(m)}Q_N^{(n)*}Q_{k+1}^{(n)*}\right]. \quad (A.6)$$

Defining $[\mathbf{R}_p]_{m,n} \equiv Q_m^{(p)}Q_n^{(p)}$, and using eigenvalues and eigenvectors (22,23), magnetization on site $k$ is obtained as

$$m_{k,\mathrm{L}}^z = \begin{cases} \dfrac{l\mu_{\mathrm{L}}}{l+r}\mathrm{Re}\left[\displaystyle\sum_q \dfrac{U_{N-k}\left(\frac{\tilde{\lambda}_q^*}{2}\right)+irU_{N-k-1}\left(\frac{\tilde{\lambda}_q^*}{2}\right)}{U_{N-1}\left(\frac{\tilde{\lambda}_q^*}{2}\right)}[\mathbf{R}_q^*]_{1,k}\right], & (k=1\sim N-1), \\[3em] \dfrac{l\mu_{\mathrm{L}}}{(l+r)(1+lr)}, & (k=N), \end{cases} \quad (A.7)$$

$$m_{k,\mathrm{R}}^z = \begin{cases} \dfrac{r\mu_{\mathrm{R}}}{(l+r)(1+lr)}, & (k=1), \\[2em] \dfrac{r\mu_{\mathrm{R}}}{l+r}\mathrm{Re}\left[\displaystyle\sum_q \dfrac{U_{k-1}\left(\frac{\tilde{\lambda}_q^*}{2}\right)+ilU_{k-2}\left(\frac{\tilde{\lambda}_q^*}{2}\right)}{U_{N-1}\left(\frac{\tilde{\lambda}_q^*}{2}\right)}[\mathbf{R}_q^*]_{N,k}\right], & (k=2\sim N), \end{cases} \quad (A.8)$$

and spin current between sites $k$ and $k+1$ is obtained as

$$j^z_{k,k+1,\mathrm{L}} = \frac{4Jl\mu_{\mathrm{L}}}{l+r}\,\mathrm{Im}\left[\sum_q \frac{U_{N-k}\left(\frac{\tilde{\lambda}^*_q}{2}\right)+irU_{N-k-1}\left(\frac{\tilde{\lambda}^*_q}{2}\right)}{U_{N-1}\left(\frac{\tilde{\lambda}^*_q}{2}\right)}\left[\mathbf{R}^*_q\right]_{1,k+1}\right], \tag{A.9}$$

$$j^z_{k,k+1,\mathrm{R}} = \begin{cases} -\dfrac{4Jlr\mu_{\mathrm{R}}}{(l+r)(1+lr)}, & (k=1), \\[2ex] \dfrac{4Jr\mu_{\mathrm{R}}}{l+r}\,\mathrm{Im}\left[\displaystyle\sum_q \frac{U_{k-1}\left(\frac{\tilde{\lambda}^*_q}{2}\right)+ilU_{k-2}\left(\frac{\tilde{\lambda}^*_q}{2}\right)}{U_{N-1}\left(\frac{\tilde{\lambda}^*_q}{2}\right)}\left[\mathbf{R}^*_q\right]_{N,k+1}\right], & (k=2\sim N), \end{cases} \tag{A.10}$$

where the parameters $l, r$ are defined below (24) and $U_k(x)$ is Chebyshev polynomial of the second kind for order $k$. Calculating these formulas, we derive the following Lemma.

**Lemma 2.** For the Hermitian conjugate of normalized matrix $\tilde{\Xi} \equiv (\Xi - B\mathbb{1})/J$, the component of $(k-m)$-th power of the normalized matrix $\tilde{\Xi}$ is obtained as

$$\left[(\tilde{\Xi}^\dagger)^{k-m}\right]_{1,k} = \begin{cases} il, & (m=0), \\ 1, & (m=1), \\ 0, & (m=2\sim k). \end{cases} \tag{A.11}$$

This lemma can be proved easily. Since the normalized matrix $\tilde{\Xi}^\dagger$ has non-zero term at only secondary-diagonal part, the $(1,k)$-component of $(k-m)$-th power of the normalized matrix $\tilde{\Xi}$ is

$$\left[(\tilde{\Xi}^\dagger)^{k-m}\right]_{1,k} = \tilde{\Xi}^\dagger_{1,m+1}\tilde{\Xi}^\dagger_{m+1,m+2}\tilde{\Xi}^\dagger_{m+2,m+3}\cdots\tilde{\Xi}^\dagger_{k-1,k}. \tag{A.12}$$

For all $m(0 \leq m \leq k)$, the component $\tilde{\Xi}^\dagger_{j,j+1}$ is equal to 1, so the component $\left[(\tilde{\Xi}^\dagger)^{k-m}\right]_{1,k}$ is equal to $\tilde{\Xi}^\dagger_{1,m+1}$. Therefore, the component $\left[(\tilde{\Xi}^\dagger)^{k-m}\right]_{1,k}$ is classified by $\tilde{\Xi}^\dagger_{1,m+1}$.

By this lemma, magnetization and spin current for steady state is simplified. By using the recurrence relation for Chebyshev polynomial of the second kind $U_{n+1}(x) = 2xU_n(x) - U_{n-1}(x)$, the numerators in (A.7-A.10) is calculated as

$$U_{N-k}+irU_{N-k-1} = \left\{\frac{ir}{1+rl}\left(\tilde{\lambda}^*_q\right)^k + \frac{1+r(r+l)}{1+rl}\left(\tilde{\lambda}^*_q\right)^{k-1}+\mathcal{O}\!\left(\left(\tilde{\lambda}^*_q\right)^{k-2}\right)\right\}U_{N-1}, \tag{A.13}$$

$$(U_{k-1}+ilU_{k-2})(U_{N-1}-ilU_{N-2}) = \left(-\frac{il}{1+rl}\left(\tilde{\lambda}^*_q\right)^k + \frac{1}{1+rl}\left(\tilde{\lambda}^*_q\right)^{k-1}+\mathcal{O}\!\left(\left(\tilde{\lambda}^*_q\right)^{k-2}\right)\right)U_{N-1}. \tag{A.14}$$

Substituting (A.13,A.14) to (A.7-A.10),

$$m^z_{k,\mathrm{L}} = \begin{cases} \dfrac{l\mu_{\mathrm{L}}}{l+r}\,\mathrm{Re}\left[\dfrac{ir}{1+rl}\left(\tilde{\Xi}^\dagger\right)^k + \dfrac{1+r(r+l)}{1+rl}\left(\tilde{\Xi}^\dagger\right)^{k-1}+\mathcal{O}\!\left((\tilde{\Xi}^\dagger)^{k-2}\right)\right]_{1,k}, & (k=1\sim N-1), \\[2ex] \dfrac{l\mu_{\mathrm{L}}}{(l+r)(1+lr)}, & (k=N), \end{cases} \tag{A.15}$$

$$m_{k,\mathrm{R}}^z = \begin{cases} \dfrac{r\mu_{\mathrm{R}}}{(l+r)(1+lr)}, & (k=1), \\ \dfrac{r\mu_{\mathrm{R}}}{l+r}\,\mathrm{Re}\left[-\dfrac{il}{1+rl}\left(\tilde{\Xi}^\dagger\right)^k + \dfrac{1}{1+rl}\left(\tilde{\Xi}^\dagger\right)^{k-1} + \mathcal{O}((\tilde{\Xi}^\dagger)^{k-2})\right]_{1,k}, & (k=2\sim N), \end{cases}$$ (A.16)

$$j_{k,k+1,\mathrm{L}}^z = \dfrac{4Jl\mu_{\mathrm{L}}}{l+r}\,\mathrm{Im}\left[\dfrac{ir}{1+rl}\left(\tilde{\Xi}^\dagger\right)^k + \dfrac{1+r(r+l)}{1+rl}\left(\tilde{\Xi}^\dagger\right)^{k-1} + \mathcal{O}((\tilde{\Xi}^\dagger)^{k-2})\right]_{1,k+1},$$ (A.17)

$$j_{k,k+1,\mathrm{R}}^z = \begin{cases} -\dfrac{4Jlr\mu_{\mathrm{R}}}{(l+r)(1+lr)}, & (k=1), \\ \dfrac{4Jr\mu_{\mathrm{R}}}{l+r}\,\mathrm{Im}\left[-\dfrac{il}{1+rl}\left(\tilde{\Xi}^\dagger\right)^k + \dfrac{1}{1+rl}\left(\tilde{\Xi}^\dagger\right)^{k-1} + \mathcal{O}((\tilde{\Xi}^\dagger)^{k-2})\right]_{1,k+1}, & (k=2\sim N). \end{cases}$$ (A.18)

Applying lemma to the above formulas, the magnetization and spin current in NESS can be expressed in terms of model parameters as follows,

$$m_k^z = \mu_{\mathrm{L}} - \dfrac{j}{4J}D_k^{(\mathrm{L})} = \mu_{\mathrm{R}} + \dfrac{j}{4J}D_k^{(\mathrm{R})}, \quad j = \dfrac{\varepsilon_{\mathrm{L}}\varepsilon_{\mathrm{R}}(\mu_{\mathrm{L}} - \mu_{\mathrm{R}})}{4J\left(1 + \frac{\varepsilon_{\mathrm{L}}}{4J}\frac{\varepsilon_{\mathrm{R}}}{4J}\right)\left(\frac{\varepsilon_{\mathrm{L}}}{4J} + \frac{\varepsilon_{\mathrm{R}}}{4J}\right)}.$$ (A.19)

The sequences $D^{\mathrm{L/R}}$ are defined as

$$D_k^{(\mathrm{L})} = \left\{\dfrac{4J}{\varepsilon_{\mathrm{L}}}, \dfrac{\varepsilon_{\mathrm{L}}}{4J} + \dfrac{4J}{\varepsilon_{\mathrm{L}}}, \cdots, \dfrac{\varepsilon_{\mathrm{L}}}{4J} + \dfrac{4J}{\varepsilon_{\mathrm{L}}}, \dfrac{\varepsilon_{\mathrm{L}}}{4J} + \dfrac{4J}{\varepsilon_{\mathrm{L}}} + \dfrac{\varepsilon_{\mathrm{R}}}{4J}\right\},$$ (A.20)

$$D_k^{(\mathrm{R})} = \left\{\dfrac{\varepsilon_{\mathrm{R}}}{4J} + \dfrac{4J}{\varepsilon_{\mathrm{R}}} + \dfrac{\varepsilon_{\mathrm{L}}}{4J}, \dfrac{\varepsilon_{\mathrm{R}}}{4J} + \dfrac{4J}{\varepsilon_{\mathrm{R}}}, \cdots, \dfrac{\varepsilon_{\mathrm{R}}}{4J} + \dfrac{4J}{\varepsilon_{\mathrm{R}}}, \dfrac{4J}{\varepsilon_{\mathrm{R}}}\right\}.$$ (A.21)

## B  Calculation of time derivative of magnetization

In this appendix, we study large $N$ behavior of $\sum_{n=1}^N e^{-2it\lambda^{(n)}}Q_j^{(n)}Q_k^{(n)}$ which appears in the expression of $\mu_k(t)$ in (55) and derive the integral formula (56). We also study some of its properties. First we divide the sum over $n$ into two parts corresponding to normal eigenstates and special eigenstates as

$$\sum_{n=1}^N e^{-2t\beta_n}Q_j^{(n)}Q_k^{(n)} = \sum_{n\in\{\mathrm{no}\}} e^{-2t\beta_n}Q_j^{(n)}Q_k^{(n)} + \sum_{n\in\{\mathrm{sp}\}} e^{-2t\beta_n}Q_j^{(n)}Q_k^{(n)},$$ (B.1)

where $\beta_n = i\lambda^{(n)}$ and $\{\mathrm{no}\} = \{1, 2, \cdots, N\}\setminus\{\mathrm{sp}\}$. For large $N$, the normalization factor $\mathcal{N}_n$ for normal eigenstate, defined below (29), can be calculated using the component of the $l$-th eigenvector corresponding to a normal eigenvalue (23) as

$$\mathcal{N}_n^2 \approx \dfrac{N}{2\sin^2\theta_n}\left(1 + 2il\cos\theta_n - l^2\right),$$ (B.2)

where the parameter $l$ is defined below (24).

Using $\beta_n = 2iJ\cos\frac{n}{N+1}\pi + \mathcal{O}(N^{-2})$ and (47), the summation can be calculated as

$$\sum_{n\in\{\mathrm{no}\}} e^{-2t\beta_n}Q_j^{(n)}Q_k^{(n)}$$

$$\approx \dfrac{2}{\pi}\int_0^\pi \dfrac{e^{-4iJt\cos x}}{1 + 2il\cos x - l^2}(\sin jx + il\sin(j-1)x)(\sin kx + il\sin(k-1)x)\,\mathrm{d}x$$

$$= \oint_C \dfrac{\mathrm{d}z}{2\pi i}e^{2Jt(z-z^{-1})}\left\{\dfrac{i^{k-j}}{z^{k-j-1}} + \dfrac{i^{j+k}(z+l)z^{j+k-2}}{lz-1}\right\},$$ (B.3)

where in the last expression the contour $C$ is the unit circle around the origin.

As for the contributions from the special eigenvalues, one can see that the normalization behaves as

$$\mathcal{N}_{\text{sp}}^2 \approx \begin{cases} \left(1+l^{-2}\right)^{-1}, & (l>1), \\ \dfrac{(l-r)^2\,(-ir)^{2N+2}}{\left(1+r^2\right)^3}, & (r>1). \end{cases} \tag{B.4}$$

The leading term for the part of the special eigenstates is calculated as

$$e^{-2t\lambda_{\text{sp}}}Q_j^{(\text{sp})}Q_k^{(\text{sp})} \approx \begin{cases} -e^{-2J\left(l-l^{-1}\right)t}\left(1+l^2\right)(-il)^{-j-k}, & (l>1), \\ -e^{-2J\left(r-r^{-1}\right)t}\left(1+r^2\right)(-ir)^{-2N-2+j+k}, & (r>1). \end{cases} \tag{B.5}$$

The two contributions, (B.3) and (B.5), can be combined into a single contour integral formula (56) by taking the contour $C$ as described. By setting $j=k$ in (B.5) we find $|Q_j^{(\text{sp})}|^2 \approx (1+l^2)l^{-2j}$ when $l>1$, implying that a special eigenstate is a mode localized at the boundary and has a decay correlation length $1/(2\log l)$ (the same argument can be applied for $r>1$ as well). As we will show below the special eigenstates do not give particular contributions for quantities studied in this paper.

Expanding the integrand in powers of $l$ (when $l<1$, or in powers of $1/l$ when $|l|>1$) and using the integral form of the Bessel function of $n$th order

$$J_n(z) = \frac{i^n}{\pi}\int_0^\pi e^{-iz\cos\theta}\cos n\theta\,d\theta, \tag{B.6}$$

an alternative formulas for $f(j,k;t)$ in terms of Bessel functions are found. They are summarized as follows and are useful for numerical evaluations:

$$f_{\text{no}}(j,k;t) = \begin{cases} (-1)^{k+1}J_{j-k}(4Jt)-J_{j+k-2}(4Jt) \\ \quad +\sum\limits_{n=0}^\infty (-l)^n\left(J_{j+k+n-2}(4Jt)+J_{j+k+n}(4Jt)\right), & (\varepsilon_{\text{L}}<4J), \\ Z_{j,k}(4Jt)+(-1)^{j+k+1}, & (\varepsilon_{\text{L}}=4J), \\ (-1)^k J_{j-k}(4Jt)-J_{j+k}(4Jt) \\ \quad +\sum\limits_{p=0}^\infty (-l)^{-p}\left(J_{j+k-p-2}(4Jt)+J_{j+k-n}(4Jt)\right), & (\varepsilon_{\text{L}}>4J), \end{cases} \tag{B.7}$$

$$f_{\text{sp}}(j,k;t) \equiv e^{-2J\left(l-l^{-1}\right)t}\left(1+l^2\right)l^{-j-k}I(l)+(-1)^{N+1}e^{-2J\left(r-r^{-1}\right)t}\left(1+r^2\right)r^{-2N-2+j+k}I(r), \tag{B.8}$$

where the function $I(x)$ takes the value 1 if $x>1$ and 0 if $x\le 1$ and the function $Z_{j,k}(4Jt)$ is defined as,

$$Z_{j,k}(4Jt) = \begin{cases} (-1)^k J_{j-k-2}(4Jt)-J_{j+k-1}(4Jt)+2\sum\limits_{n=0}^\infty (-1)^n J_{j+k+n}(4Jt), & (j>k), \\ -J_{2k-1}+2\sum\limits_{n=0}^\infty \left\{(-1)^k J_{2n+2}(4Jt)+(-1)^n J_{2k+n}(4Jt)\right\}, & (j=k). \end{cases} \tag{B.9}$$

Next we will see that $j=1$ case of (56), i.e.,

$$f(1,k;t) = \oint_C \frac{dz}{2\pi i}\,e^{2Jt\left(z-z^{-1}\right)}\frac{i^{k+1}(z^k+z^{k-2})}{lz-1}, \tag{B.10}$$

is close to zero except near $t \sim k/(4J)$. For large $t$, we may use the saddle point analysis with $t = \alpha k$. Let us first write

$$f(1, k; t) = \oint_C \frac{dz}{2\pi} g(z) e^{kf(z)}, \tag{B.11}$$

with

$$f(z) = 2J\alpha(z - 1/z) + \log z + \frac{i\pi}{2}, \quad g(z) = \frac{1 + z^{-2}}{lz - 1}. \tag{B.12}$$

It is easy to check that the two roots of $f'(z) = 0$ are given by

$$z = -\frac{1}{4J\alpha} \pm \sqrt{\frac{1}{16J^2\alpha^2} - 1} =: z_\pm. \tag{B.13}$$

When $0 < \alpha < 1/4J$, the saddle point is at $z = z_+$ and we find

$$f(1, k; t) \sim (2\pi)^{-1/2}(1 - 16J^2\alpha^2)^{-1/4} \frac{1 + z_+^{-2}}{2\pi(lz_+ - 1)}(iz_+)^{k+1} e^{k\sqrt{1 - 16J^2\alpha^2}}, \tag{B.14}$$

and hence

$$|f(1, k; t)| \sim (2\pi)^{-1/2}(1 - 16J^2\alpha^2)^{-1/4} \left| \frac{1 + z_+^{-2}}{2\pi(lz_+ - 1)} z_+^{k+1} \right| e^{k\sqrt{1 - 16J^2\alpha^2}}. \tag{B.15}$$

On the other hand, when $\alpha > 1/4J$, two saddle points are at the unit circle $(z_\pm = e^{\pm i\theta})$ and we find

$$f(1, k; t) \sim \frac{2(2\pi k)^{-1/2}(16J^2\alpha^2 - 1)^{-1/4}}{1 - 2l\cos\theta + l^2} i^k \mathrm{Im}\left[(1 + e^{-2i\theta})(1 + le^{-i\theta})e^{ik\sqrt{16J^2\alpha^2 - 1} + i(k+1)\theta + i\pi/4}\right], \tag{B.16}$$

and hence

$$|f(1, k; t)| \sim \frac{(2\pi k)^{-1/2}(16J^2\alpha^2 - 1)^{-1/4}\sqrt{1 - l/2J\alpha + l^2}}{(1 + l/2J\alpha + l^2)} |\sin[k(\sqrt{16J^2\alpha^2 - 1} + \theta) + \phi]|, \tag{B.17}$$

where

$$\tan\phi = \frac{1 - l/4J\alpha - l\sqrt{1 - 1/16J^2\alpha^2}}{1 - l/4J\alpha + l\sqrt{1 - 1/16J^2\alpha^2}}, \tag{B.18}$$

and in the last equality we used $\cos\theta = -1/4J\alpha$. These asymptotic behaviors indicate that the function $f(1, k; t)$ becomes quickly small when $t < k/(4J)$ and shows oscillatory decay when $t > k/(4J)$. The expressions above diverge when $\alpha \to 1/4J$ but this may be remedied by noting that the saddle point becomes degenerate and one has to use a different asymptotics. For small $t$ and fixed $k$, we may also discuss as follows. First expand (B.10) in powers of $t$. When $|l| < 1$, we get

$$f(1, k; t) = \sum_{n,p=0} \frac{(-1)^{n+p+k-1}(2Jt)^{2n+p+k-1}}{n!(n+p+k-1)!} l^p + \sum_{n,p=0} \frac{(-1)^{n+p+k+1}(2Jt)^{2n+p+k+1}}{n!(n+p+k+1)!} l^p. \tag{B.19}$$

The leading terms for small $t$ are when $n = p = 0$ and

$$f(1, k; t) \approx \frac{(-2Jt)^{k-1}}{(k-1)!} + \frac{(-2Jt)^{k+1}}{(k+1)!}. \tag{B.20}$$

By the Stirling formula, these terms are small when $t < k/(4J)$. We can find a similar expansion also when $|l| > 1$ and come to the same conclusion that it is small when $t < k/(4J)$.

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
