# Peer review of "Exact solution for the Lindbladian dynamics for the open XX spin chain with boundary dissipation"

_SciPost Physics, doi:SciPost Phys. 14, 112 (2023)_

## Round 1 · Referee Report · Anonymous · 2021-6-3

Strengths
1 - technical novelty
2 - derivation of novel exact results
Weaknesses
1 - the phenomena described in this paper are well known, or could have been at least
Report
In this paper, the authors provide exact formulas describing how single particle (quadratic) physical observables of an XX spin chain (or free fermion system) approach their stationary value after the sudden connection to two biased reservoirs, which are described in the Lindblad formalism. This kind of setup is currently actively studied as it as paved the way to important discoveries concerning the exotic transport properties of integrable quantum systems.
The main interest of this work is mainly technical. The authors show that the standard third-quantization procedure, which is normally used to analyze these systems, can be considerably simplified and brought to the diagonalization of NxN matrices instead of 2Nx2N, where N is the number of lattice sites of the spin-chain.
Now, the authors should clarify a little bit their claims of novelty here. For instance also in references [56-57], the authors of those papers claim to have obtained an analogous 2Nx2N -> NxN reduction and to rely on that to find exactly the spectrum of the Lindbladian (but only under limited assumptions and for some specific choices of parameters). I am convinced that the method presented by the authors in this manuscript is way more general and powerful, but still, it would be appreciable if the authors could make more clear the degree of novelty of their work and results, in the Introduction and throughout the text, compared to previous works. This would avoid confusion for the readers. In other words, what was done here that could not be done with the formalism developed in e.g. [56-57] ?
The authors rely then on their formalism to inspect the out-of-equilibrium dynamics of the magnetization after an XX chain prepared in the infinite temperature state is connected to two biased reservoirs. They spend particular attention concerning the time-scales at which the stationary value is attained at the bulk and close to the boundary, which are different. Then, they deeply analyze a plateau regime for the magnetization (Fig. 6c), which persists until the perturbation caused by the the opposite boundary propagates until it reaches the monitored sites. This interpretation follows directly from the light-cone dynamics presented mainly in Fig. 3, which is expected for such a non-interacting system where free excitations propagate ballistically.
I do not find the phenomenology presented in the paper particularly surprising or original, but its exact description required serious technical effort and I definitively recommend the paper to be published in its current form, after minor modifications.
I wonder if the authors considered or find themselves in position to extend their results to the case of realistic reservoirs at some finite temperature? In that case the dynamics is still described by a quadratic theory and should be in principle solvable. It may be interesting to investigate the real-time approach to a thermal state at finite temperature starting from an infinite temperature one.
Requested changes
1 - clarify degree of technical novelty in the main text, in particular compared to Refs [56-57]
2-Correct minor typos/modify sentences :
- page 2 : "we obtain the exact solutions for NESS" -> we obtain the exact solutions for the NESS ?
- page 2 : "as a site gets close to the center of the chain, the plateau regime is quickly short by a fluctuation and the convergence to the steady state value is slow." si "quickly short" what the authors actually meant?
- page 5 : "This Hamiltonian in (1)" -> The Hamiltonian in (1) ?
- page 20: "find the emergence of a special eigenvalue which is a larger imaginary part than the"
-> "which has a larger" ?
- page 21 : "beyond the post-selection which usually approximates in the context of non-Hermitian physics" I am not sure to understand this sentence.
Author: Kohei Yamanaka on 2021-07-11 [id 1563]
(in reply to Report 1 on 2021-06-03)
We thank the referee for the useful suggestions. We have improved our paper by implementing all of them. About an extension to the case of realistic reservoirs at some finite temperature, we have not considered it carefully yet. But it is certainly an interesting direction and we plan to study it.
Referee writes:
clarify the degree of technical novelty in the main text, in particular, compared to Refs [56-57]
Our comments: We have clarified technical novelty in our paper. For refs[56-57], in page 3, we have added the remarks that the specific cases, such as the open XX spin chain whose specific dissipative strengths satisfy the certain condition, and the open XY spin chain without magnetic field, are calculated in the previous works [56,57]. Also, in order to make the novelty of our study clear, we have emphasized that our procedure can be applied to a general situation in page 6.
Moreover, in page 10, we have added a reason why the equation (37) is not solved in previous papers. In the papers [56,57], the eigenvectors are not exactly obtained.
Referee writes:
Correct minor typos/modify sentences :
Our comments: All noticed typos have been fixed.
All noticed modifications of sentences are improved as follows:
-
page 3: "as a site gets close to the center of the chain, the plateau regime is quickly short by a fluctuation and the convergence to the steady state value is slow." ->"The plateau regime at a site in the bulk but not at the center has a finite duration but becomes rapidly short by a fluctuation as it gets close to the center"
-
page 21: the sentences including "beyond the post-selection which usually approximates in the context of non-Hermitian physics" ->"Recently, the analysis of non-Hermitian systems have been applied to open quantum systems by using the post-selection [81,82], and many interesting properties for open quantum systems, such as phase transitions [83] and topological natures [84], have been studied. However, it has been known that the dynamics which is described by the non-Hermitian systems is different from the Lindblad dynamics [81,82]. We hope that our exact results will be useful for future studies of systems described the Lindblad equation."
Author: Kohei Yamanaka on 2022-12-22 [id 3173]
(in reply to Report 2 on 2021-07-16)First of all we would like to thank the referee for his/her careful reading of our manuscript and for giving many useful remarks. Second we apologize for its taking so long to revise the paper. There are several reasons for it; one is that we realized that the revision should be substantial (as explained below), and another is that the first author could not work on the paper for a fairy long time because he was involved in activities to find a job after finishing PhD.
We have carefully considered the comments by the referee. The main changes we have made in the revised version are summarized as follows.
We have carefully reconsidered the long time behaviors of physical quantities at various positions in the system. In the original version of the paper, the analysis of the Liouvillian gap was based on numerical estimates of our formula for finite N and the position dependence of the long time behaviors were not clearly explained, as pointed out by the referee. In the revised version we examined the large N behavior of the Liouvillian gap more in detail analytically and were able to determine the coefficient in front of the 1/N^3 scaling.
During the process of this reexamination of the long time behaviors, we realized that the apparent special behaviors near the center of the chain may be a numerical artifact due to the absence of the plateau region at the center. Now we believe that in fact the long time behaviors of physical quantities are simply described by the Liouvillian gap at an arbitrary site. On the other hand we have also performed careful asymptotic analysis and have found much clearer understanding of the plateau regime. Difference of behaviors in this regime between sites near the boundaries and in the bulk are better explained in the revised version. But as specialities of the center of the chain, we now mention only the vanishing plateau region but do not claim the long time behaviors. We apologize for the misleading explanations on this point in the original version.
Below we reply to each item in the comments more concretely. (We upload a version of our manuscript in which the improved parts are displayed in red so that you can find them easily. There are also parts in orange corresponding to the other referee.)
<Requested changes> The referee writes: - The authors should celery define the quantity plotted in Fig. 4(b).
Our response: The figure in question is removed in the revised version.
The referee writes: - Inset a figure similar to Fig.(4) for a site in the middle of the chain.
Our response: We are not sure if “Fig. (4)” was correct because the figure for the middle of the chain was provided in the original version. Because we do not claim special behaviors near the center, the figure for the center is not included in the revised version.
The referee writes: - Add a discussion of the finite-size effects of the position-dependent decay rate with N
Our response: We have performed large N analysis of the Liouvillian gap in section 4.3. We have carefully analyzed the finite-size effects of the position-dependent decay rate with N by following the suggestion by the referee, but we realized that there is not really a special behaviors near the center and decided not to include the discussion on this. On the other hand, position dependence in the plateau region is now better understood and is explained in section 4.2.
<Main criticism> The referee writes: - I believe the authors do now explore the full power of their analytical solution.
Our response: We agree with this comment and have performed more careful analytical studies in detail.
The referee writes: -Besides the decay factor, the overlap should have an N-dependent prefactor. Can this be obtained analytically?
Our response: -We have performed large N analysis of the Liouvillian gap and could determine its coefficient in section 4.3. We regard this as an important conclusion from our exact analytical solution.
The referee writes: - A particularly ... - The authors should study ....
Our response: As already explained above, the slow decay only near the center of the chain, which we stressed in the original version, was in fact an artifact due to the vanishing plateau regime there. We have examined this point carefully and decided not to include discussions on this in the revised version.
<Other points> The referee writes: - The authors reproduce the result predicting the existence of one or two eigenvalues isolated from...
Our response: One or two eigenvalues isolated from the rest of the spectrum, which we call the special eigenvalues in the paper, are indeed localized near the boundaries and their localization length can be calculated using our exact solution. Unfortunately they do not give meaningful effects to behaviors of physical quantities studied in this paper (it would be interesting to find physically meaningful effects of these modes). Discussions about these are included in Appendix B in the revised version.
<Weakness> The referee writes: - Written in a rather technical way. - Most findings are expected. <Report> The referee writes: - Regarding the clarity…
Our response: We agree that our paper is rather technical and also the writing is. To some extent this is on purpose because we wanted to explain clearly that Lindblad time evolution for XX spin chain with boundary dissipation can be studied quite in detail exactly. Of course many parts of this type of analysis have been already done in previous works but full time evolution of physical quantities have not been calculated in literature and we believed that it would be useful to have such formulas.
On the other hand, we have also tried to add more explanations and/or simply some parts of arguments so that our paper is more accessible to a wider range of readers.
The referee writes: - Regarding novelty. The gain of 2Nx2N->NxN was also known.
Our response: We guess the referee mentions what has been known from the connection to the Kitaev model. Indeed the mechanism of the reductions of matrix sizes seem the same and we now mention it in introduction.
Attachment:
paper1_Exactsolution-revise.pdf

---

## Round 1 · Referee Report · Anonymous · 2021-7-16

Strengths
Reports some interesting effects
Weaknesses
Written in a rather technical way
Most findings are expected
Report
This is a rather technical paper that presents an exact solution to the boundary-driven XX chain and analyses the dynamics of the system starting from a fully mixed state until attaining the steady state.
The topic is timely- open quantum systems, in particular boundary-driven models, have yielded important contributions to understand incoherent transport, relaxation, and decoherence, and their dynamics is still much less known than they of their closed counterparts.
I believe the results reported in the paper are sound and integrate with previously known results in the literature.
The first part of the paper is not very easy to read due to its technical nature. However, the results given in the second part are adequately explained.
Regarding novelty.
- The methods and the solution were known previously.
- The gain of 2Nx2N -> NxN was also known and can be simply understood from the complex fermion (rather than the Majorana fermion) perspective. Written in a complex fermion base, the XX model has now anomalous (superconducting) terms and thus is diagonalized with no requirements to the anomalous sector. This is no longer the case for the XY model.
- The detailed analysis of the time evolution is novel and presents some interesting aspects:
— The identification of the light cone is expected but had not been observed in this way.
— The convergence arising in plateaus is also not unexpected. This kind of behavior is seen in the relaxational dynamics of local observables in closed 1d systems with open boundary conditions. However, it is here reported for the dissipative case which makes it qualitatively different from previously reported accounts.
— The identification that relaxation of observable in the middle or near the end of the chain is different is novel and an interesting finding of this work.
Regarding the clarity and organization of the paper.
- I believe most of the results in the paper can be explained to a reader not familiar with the technical details. However, the way the authors intercalate the technical and discussion parts makes it difficult to pass the paper’s main messages to a non-technical oriented audience. Solving this would require substantial rearrangements of the text and is ultimately a question of style. So although I would suggest the authors make the manuscript more reader-friendly, this is not a requrement.
Main criticism.
I believe the authors do not explore the full power of their analytical solution.
A particularly interesting question, in my view, is the remark that a site in the middle of the chain relaxes shower to the steady-state then one near the boundary.
This statement is made under rather imprecise assumptions. The authors should provide quantitative and qualitative statements which I believe their exact solution may provide.
The authors should study the finite-size scaling with N of the decay of modes in the middle of near the edges.
Fig.4(b) [see requested changes] shows that the decay of the overlap as a function of the distance to be boundaries is exponential suppressed. This means there is a correlation length, that should be independent of N but dependent on the coupling to the environment. Can the authors obtain this behavior from your analytical solution?
Besides the decay factor, the overlap should have an N-dependent prefactor. Can this be obtained analytically?
Can these two last results explain the finite-size scaling?
Other points.
The authors reproduce the result predicting the existence of one or two eigenvalues isolated from the rest of the spectrum for high-enough system-environment coupling.
These seem to be highly dissipative eigenmodes, which, I believe, should be localized near the boundaries.
From their exact solutions can the authors give more insight about such modes, in particular their localization length?
Are there and physical consequences of these states for the dynamics or any steady-state observables?
Assessment.
I can recommend the manuscript for publication after the authors successfully reply to my criticism and/or make the necessary changes.
Requested changes
- The authors should clearly define the quantity plotted in Fig.4(b). I believe it is the amplitude on site of the eigenmode of X with the smallest imaginary part, but I’m not sure.
- Insert a figure similar to Fig.(4) for a site in the middle of the chain.
- Add a discussion of the finite-size effects of the position-dependent decay rate with N

---

## Round 2 · Referee Report · Anonymous (Referee 2) · 2023-2-11

Strengths

Sound calculations compared with numerics

Weaknesses

Written in a rather technical way
Most findings are expected

Report

In the revised version the authors (i) develop a method for analytically determining the post-quench dynamics of the magnetisation and spin current of the boundary-driven XX chain. Subsequently, the authors use their method to obtain two concrete results (ii) the height of the magnetisation plateau observed in intermediate-time dynamics, and (iii) the coefficient of the long-time relaxation rate to the steady-state. This revised version of the manuscript is significantly different from the previous one, having in common only the result (i).

Regarding novelty

I maintain my main view. Despite not being particularly surprising or original, the phenomenology presented in the paper requires serious technical effort.

Results (i) are relatively straightforward, but technically involved, and are useful to other researchers looking at analytical aspects of post-quensch dynamics in open systems.

Results (ii) and (iii) are novel. They also seem to be sound and were tested against numerical simulations.

Regarding the clarity and organization of the paper.

I maintain my previous viewpoint. There was no substantial change in this aspect.

Main criticism.

Most of my previous criticisms no longer apply since the paper was substantially modified.

I believe that the authors now explore the power of their analytical solution to obtain results (ii) and (iii). Therefore, my previous criticism was addressed.

The main critiques I have now are related to the presentation since I still find the paper hard to read. For example, it would be helpful to have the final expression of $\Delta$ in Eq.(61) terms of physical quantities $\varepsilon_{R/L}$, $J$, etc. However, to understand their formula, the reader must carefully examine the calculations to find the definitions of $l$ and $r$ below Eq (51). Putting the final result in terms of the physical quantities defined at the beginning of the manuscript could highlight physical aspects and help a non-technically inclined reader to still be able to find some useful information. As another example, b defined below Eq (51) is never used thereafter. This gives the impression that the authors passed the calculations from their notebook to the paper without thinking about the most effective way of presenting them to the reader.

Other points:

I believe there is a problem with the caption of Fig 4. Eq (59) should refer to Eq (58)

Assessment.
I recommend the manuscript for publication. However, I strongly advise the authors to improve the readability of the manuscript.
  • validity: high
  • significance: ok
  • originality: ok
  • clarity: ok
  • formatting: acceptable
  • grammar: acceptable

Author:  Kohei Yamanaka  on 2023-02-17  [id 3363]

(in reply to Report 2 on 2023-02-11)

We thank the referee for his/her careful reading of our manuscript and for giving useful comments. We have carefully checked all the comments by the referee on our manuscript and improved it. A version of our new draft, in which all changes are displayed in red, is also attached.

The referee wrote:
<Weakness>
-Written in a rather technical way
<Regarding the clarity and organization of the paper>
-I maintain my previous viewpoint. There was no substantial change in this aspect.
<Main criticism>
-The main critiques I have now are related to the presentation since I still find the paper hard to read.

Our response:
We have improved several points in our manuscript in order to make our paper more accessible to non-technically inclined readers.

First, the referee wrote:
-For example, it would be helpful to have the final expression of Delta in Eq.(61) terms of physical quantities…

Our response:
In the revised version, we follow the advice and write the final expression of $\Delta$ (61) in terms of the physical quantities $J$, $\varepsilon_L$, and $\varepsilon_R$. At the same time we also made the use of parameters l and r in a more systematic way. Namely, we now define the parameters l and r below Eq.(24) on page 7 where they appear for the first time and then consistently use them except for the final results. We believe that these modifications have simplified several formulas and allowed the final results to be understood by non-technically inclined readers.

Second, the referee wrote:
-As another example, b defined below Eq (51) is never used thereafter.

Our response:
We have rewritten b to j/(4J).

In addition to the above two points, which were pointed out by the referee, we have added a few more explanations about motivations and summaries of our discussions in sections 2 and 3 on pages 4,6, 9, and 11. By this revision, we believe that non-technical inclined readers may understand our aims and results better without reading detailed calculations.

Finally we have removed the following unnecessary abbreviations.

page 2 NEGF and QME

page 8 NE and SE

We hope that this small change also improves slightly readability of our paper.

Other points:
The referee wrote:
-I believe there is a problem with the caption of Fig 4. Eq (59) should refer to Eq (58).

Our response:
We thank the referee for pointing this out. We have corrected this typo as (59)->(58).

Additional changes in this revision.

In Fig. 1, we have rewritten the legends for each line from epsl and epsr to the Greek letters $\varepsilon_L$ and $\varepsilon_R$.

In Eq.(23) we set the numerator of the first factor $q_1^{(k)}$ to unity because we can choose $q_1^{(k)}$ to be an arbitrarily non-zero real number and it would be easier to understand the final result.

Attachment:

paper1_Exactsolution-reviselatest.pdf

---

## Round 2 · Referee Report · Anonymous (Referee 1) · 2023-2-13

Strengths

1 - technical novelty
2 - derivation of novel exact results

Weaknesses

1 - the phenomena described in this paper are unsurprising and can be expected based on very simple physical arguments

Report

After revision, the authors have extensively modified their manuscript, in particular starting from page 14. They have removed some figures (without loss of clarity) and also made the physical interpretation of their results more clear.

They have also clarified their statements concerning the degree of novelty of their technical results.

Requested changes

Minor comments/typos:

Typo Eq. 4? \varepsilon_R -> \varepsilon_L in L_2 ?

Same in Eq. 8 ?

Typo page 13 : Lindbradian -> Lindbladian

  • validity: high
  • significance: ok
  • originality: good
  • clarity: high
  • formatting: good
  • grammar: acceptable

Author:  Kohei Yamanaka  on 2023-02-17  [id 3364]

(in reply to Report 1 on 2023-02-13)

We thank the referee for his/her careful reading of our manuscript and for giving useful comments. The only requested changes were a few minor comments/typos. All noticed typos and comments have been fixed.

---

## Round 2 · Author Response

Dear editors and referees,

We have revised the manuscript "Exact solution for the Lindbladian dynamics for the open XX spin chain with boundary dissipation" according to the referees' comments. We believe we have been able to answer to almost all the comments and to produce an improved version of the paper. We apologize that it has taken so long to revise the paper.

---

## Round 2 · List of Changes

• We clarified technical novelty compared to previous studies in section 2.

• We completely rewrote section 4. Two main changes are the followings.

• We added detailed discussion of the plateau regime by performing an asymptotic analysis of integral representations of physical quantities.

• We examined the large N behavior of the Liouvillian gap more in detail analytically and were able to determine the coefficient in front of the 1/N^3 scaling.

• We correctted minor typos/modify sentences.

---

## Round 3 · Author Response

Dear editors and referees,

We have revised the manuscript "Exact solution for the Lindbladian dynamics for the open XX spin chain with boundary dissipation" according to the referees' comments. We believe we have been able to answer to almost all the comments and to produce an improved version of the paper.

---

## Round 3 · List of Changes

•We have improved the following several points in our manuscript in order to make our paper more accessible to non-technically inclined readers.

•We wrote the final results in terms of the model parameters to be understood by non-technically inclined readers.

•We added a few more explanations about motivations and summaries of our discussions.

•We removed the unnecessary abbreviations and definitions.

• We corrected minor typos/modify sentences.

---

## Editorial Decision

published